

# Model simulation of ammonium and nitrate aerosols distribution in the Euro-Mediterranean region and their radiative and climatic effects over 1979-2016

Thomas Drugé[1], Pierre Nabat[1], Marc Mallet[1], and Samuel Somot[1]

[1]CNRM, Université de Toulouse, Météo-France, CNRS, Toulouse, France

**Correspondence:** T. Drugé (thomas.druge@meteo.fr)

**Abstract.** Aerosols play an important role in Europe and the Mediterranean area where different sources of natural and anthropogenic particles are present. Among them ammonium and nitrate (A&N) aerosols may have a growing impact on regional climate. In this study, their representation in coarse and fine modes has been introduced in the prognostic aerosol scheme of the ALADIN-Climate regional model. This new aerosol scheme is evaluated over Europe and the Mediterranean Sea, using two

twin simulations over the period 1979-2016 with and without A&N aerosols. This evaluation is performed at local and regional scales, using surface stations and satellite measurements. Despite an overestimate of the surface nitrate concentration, the model is able to reproduce its spatial pattern including local maxima (Benelux, Po valley). Concerning the simulated Aerosol Optical Depth (AOD), the inclusion of A&N aerosols significantly reduces the model bias compared to both AERONET stations and satellite data. Our results indicate that A&N aerosols can contribute up to 40% to the total $AOD_{550}$ over Europe,

with an average of 0.07 (550 nm) over the period 2001-2016. Sensitivity studies suggest that biases still present are related to uncertainties associated with the annual cycle of A&N aerosol precursors (ammonia and nitric acid). The decrease of sulphate aerosol production over Europe since 1980 produces more free ammonia in the atmosphere leading to an increase in A&N concentrations over the studied period. Analyses of the different aerosol trends have shown for the first time to our knowledge, that since 2005 over Europe, A&N $AOD_{550}$ and A&N Shortwave (SW) Direct Radiative Forcing (DRF) are found to be higher

than sulphate and organics, becoming the species with the highest AOD and the highest DRF. On average over the period 1979-2016, the A&N DRF is found to be about -1.7 W m$^{-2}$ at the surface and -1.4 W m$^{-2}$ at the Top of the Atmosphere (TOA) in all-sky conditions over Europe, with regional maxima located at the surface over the Po valley (-5 W m$^{-2}$). Finally, the dimming effect of A&N aerosols is responsible for a cooling of about -0.2°C over Europe (summer), with a maximum of -0.4°C over the Po valley. Concerning precipitations, no significant impact of A&N aerosols has been found.

*Copyright statement.*



# 1   Introduction

Aerosols are known to have an important role in the regional and global climate system because they affect cloud properties, radiative balance and the hydrological cycle (Forster et al., 2007; Tang et al., 2018). They modify the radiative budget of the Earth through different effects, which are the direct, semi-direct and indirect effects. The direct radiative effect corresponds

to the absorption and scattering of the solar incident radiation (Mitchell, 1971; Coakley Jr et al., 1983) that generally produce a cooling effect both at the surface and at the Top of the Atmosphere (TOA). The consequences of the direct radiative effect on the atmospheric dynamics is defined as the semi-direct effect (Hansen et al., 1997; Allen and Sherwood, 2010). Finally, the indirect effect represents the modification of the microphysical cloud properties, which has consequences notably on cloud albedo and lifetime (Twomey, 1977; Albrecht, 1989; Lohmann and Feichter, 2005).

Over Europe, among the different anthropogenic aerosol species, atmospheric nitrate particles represent approximatively 10-20 % of the total dry aerosol mass at the end of the 20th century (Putaud et al., 2004; Schaap et al., 2004). Ammonium and nitrate (A&N) aerosols ($NH_4NO_3$) are mainly formed by reactions in the atmosphere from ammonia ($NH_3$) and nitric acid ($HNO_3$), a photochemical product of nitric oxides ($NO_x$) oxidation (Hauglustaine et al., 2014). The most important sources of ammonia are agriculture excreta from domestic and wild animals, as well as synthetic fertilizers (Bouwman et al., 1997;

Paulot et al., 2014). Nitric acid has anthropogenic and natural sources that come mainly from fossil fuel combustion (40%), land use practices (15%) and soil emissions (10%) (Olivier et al., 1998). The $NH_3$, present in the troposphere is the main neutralizing agent for the sulphuric acid ($H_2SO_4$) and nitric acid ($HNO_3$) (Hauglustaine et al., 2014). Firstly, the $NH_3$ will react, instantaneously and irreversibly with $H_2SO_4$ to produce ammonium sulphate (($NH_4$)$HSO_4$) (Hauglustaine et al., 2014). The less abundant of the two species is the only limitation. This reaction takes priority over ammonium nitrate formation

due to the low vapour pressure of sulphuric acid. Secondly, if all $NH_3$ is consumed by the previous reactions with $H_2SO_4$, no ammonium nitrate is formed. If there is still some $NH_3$, it will neutralize the $HNO_3$ to create $NH_4NO_3$. After the small particles are in equilibrium, coarse particles of $NH_4NO_3$ are produced by heterogeneous uptake of $HNO_3$ on calcite (part of mineral dust) and sea-salt particles (Zhuang et al., 1999; Jacobson, 1999; Jordan et al., 2003; Hauglustaine et al., 2014).

Because of the difficulty to quantify the A&N effects and the high variability of these aerosols, it is difficult to simulate

their climatic effects (Bian et al., 2017). Several global climate models (GCM) have now implemented the formation of A&N aerosols (Bian et al., 2017) to quantify their impact on the present-day climate. In that context, eight global climate models from AeroCom (Aerosol Comparisons between Observations and Models) phase II project have quantified a present-day direct radiative forcing of nitrate aerosols, at the TOA, ranging from -0.12 to -0.02 W m$^{-2}$ with a mean of -0.08 $\pm$ 0.04 W m$^{-2}$ (Myhre et al., 2013). In parallel, the different climate models involved in the Atmospheric Chemistry and Climate Model Inter-

comparison Project (ACCMIP) indicate a present-day direct radiative forcing ranging from -0.03 to -0.41 W m$^{-2}$ with a mean of -0.19 $\pm$ 0.18 W m$^{-2}$ (Shindell et al., 2013). Concerning the nitrate related Aerosol Optical Depth (AOD), several studies have shown a global annual estimation ranging from 0.0023 to 0.025 at 550 nm (Bellouin et al., 2011; Shindell et al., 2013; Myhre et al., 2013; Hauglustaine et al., 2014). Many reasons could explain the significant diversity in the global simulations of nitrate concentrations among climate models. First, nitrate aerosols are involved in a very complicated chemistry and the





system sometimes cannot even be solved using the thermodynamic equilibrium approach when coarse mode, from dust and sea-salt particles, is present. Furthermore, nitrate simulation also depends on temperature, relative humidity and on various precursors such as $NH_3$, $HNO_3$, dust and sea-salt. Finally, the different global climate models account for impact on nitrate formation of dust and sea-salt very differently (Bian et al., 2017).

By the end of the twenty-first century, some climate scenarios project an increase in $NH_3$ emissions (O'Neill et al., 2016) that could offset some of the decline in $SO_2$ and $NO_x$ emissions for the anthropogenic aerosol radiative forcing (Hauglustaine et al., 2014). With the reduction in $SO_2$ emissions, less atmospheric $NH_3$ is required to neutralize the strong acid $H_2SO_4$. The excess of $NH_3$ will form A&N aerosols so that their importance is likely to increase over this century. Several global models predict an overall increase of atmospheric nitrate burden during this century based on current available emission inventories (Bauer

et al., 2007, 2016; Hauglustaine et al., 2014; Li et al., 2014). Finally, $NH_4NO_3$ will probably become the largest contributor to anthropogenic AOD by the end of this century (Hauglustaine et al., 2014). Nevertheless, the predicted trend in surface nitrate is mixed. Indeed, despite a global increase in surface nitrate, some studies estimate a surface nitrate decrease over some regional urban areas, as in North America or in the Mediterranean region, due to the decline in $NO_x$ emissions (Bauer et al., 2016; Hauglustaine et al., 2014; Trail et al., 2014).

     The Mediterranean region, with an alternating climate between hot and dry summers and mild and wet winters, is very sensitive to climate change (Nabat et al., 2016). Furthermore, this region is a crossroads of air masses carrying natural (dust, sea-salt, etc.) and anthropogenic (black carbon, sulphate, nitrate, etc.) particles. Indeed, these aerosols come from different sources around such as the Sahara desert, industries, European cities, forest fires and even the Mediterranean Sea itself (Lelieveld et al.,

2002; Nabat et al., 2013). This region is therefore very interesting to study the role of the different aerosols on the climate. However, the issue of the aerosol representation in regional climate models (RCM) has only been rarely discussed, particularly in this region, although this scale is the most adapted to the spatio-temporal variability of these aerosols (Nabat et al., 2015a). In addition, the majority of regional climate simulations carried out at the Mediterranean basin scale used a very simplified representation of aerosols not taking A&N aerosols into account (Nabat et al., 2016).

The objective of this study is to present the implementation and the evaluation of a simplified A&N module in the TACTIC aerosol scheme (Michou et al., 2015; Nabat et al., 2015a) used in the ALADIN regional climate model and to assess the direct radiative effect and climatic impact of the A&N aerosols over the Euro-Mediterranean region. After a description of the new aerosol scheme in Sect. 2 and 3, an evaluation of this scheme will be presented in Sect. 4. The radiative and climatic effects of A&N aerosols are studied in Sect. 5 before the concluding remarks in Sect. 6.

**2   Model description**

### 2.1   The ALADIN-Climate Regional Climate Model

ALADIN-Climate is a regional climate model developed at CNRM used in the present study over the Euro-Mediterranean region. It includes an interactive aerosol scheme described thereafter, and the SURFEX land surface module (Masson et al.,



2013) with the ISBA scheme (Noilhan and Mahfouf, 1996). Land surface hydrology and river flow are simulated by TRIP model (Decharme et al., 2010).

ALADIN-Climate is used here in its version 6.2 similar as in Daniel et al. (2018). It is a bi-spectral, hydrostatic limited area regional climate model with a semi-Lagrangian advection and a semi-implicit scheme. ALADIN-Climate has a 50 km horizontal resolution and 91 vertical levels. As the regional domain is not periodic, an extension zone used only for Fourier transforms has been added in order to achieve the bi-periodization. ALADIN-Climate uses the FMR shortwave (SW) radiation scheme (Fouquart and Bonnel, 1980; Morcrette et al., 2008) with 6 spectral bands, and a longwave radiation scheme (RRTM, Rapid Radiative Transfer Model) presented in Mlawer et al. (1997).

The Euro-Mediterranean domain used for this study is presented in Figure 1. It includes the official Med-CORDEX domain and has been extended to take into account the main aerosol sources potentially transported over the Euro-Mediterranean region. The domain represents 128x180 points including 11 points (North and East) for a bi-periodization zone and 8 points (on each side) for the relaxation zone. Two regions have been defined in Figure 1 (Europe and the Mediterranean Sea) for the needs of this study.

In the ALADIN-Climate model, a prognostic aerosol scheme named TACTIC (Tropospheric Aerosols for ClimaTe In CNRM-CM) has been included as presented in Michou et al. (2015) and Nabat et al. (2015a). This aerosol scheme, originally adapted from the GEMS/MACC aerosol scheme (Morcrette et al., 2009), includes up to now five aerosol types (desert dust, sea-salt, black carbon, organic matter and sulphate). These aerosols can be interactively emitted from the surface (dust, sea-salt) or from external emission data sets (black carbon, organic matter and sulphate precursors from anthropogenic and/or biomass burning emission). In the present simulations, aerosols are not included in the lateral boundary forcing because the domain is supposed to be large enough to include all the sources of aerosols affecting the Mediterranean region.

The aerosol scheme includes 12 tracers, including 11 particulate tracers and sulphate gaseous precursors ($SO_2$). For aerosol particles, 3 size bins are used for sea-salt (0.03 to 0.5, 0.5 to 5 and 5 to 20 μm), and for dust (0.01 to 1.0, 1.0 to 2.5 and 2.5 to 20 μm), 2 bins (hydrophilic and hydrophobic particles) for organics and for black carbon and 1 size bin for sulphate particles and for sulphate precursors ($SO_2$). All these aerosols are transported in the atmosphere and submitted to the dry and wet (in and below clouds) deposition. TACTIC takes into account their interactions with radiation (direct effect for all species) and the cloud albedo (first indirect effect for sulphate, organic matter and sea-salt). Aerosols do not interact with cloud microphysics (no second indirect effect). The radiative properties of each species, and notably those used as input for the radiative transfer scheme (mass extinction efficiencies, single scattering albedo and asymmetry factor) at different wavelengths and different relative humidity are set for each aerosol type following Nabat et al. (2013). Moreover, it is worth underlining that the TACTIC aerosol scheme has been designed to perform multi-decadal simulations at reasonable computation cost.

## 2.2 Implementation of the ammonium and nitrate module in TACTIC

An ammonium and nitrate (A&N) module has been recently implemented in TACTIC. This scheme was adapted from the one implemented in the INCA (Interaction with Chemistry and Aerosols) global model (Hauglustaine et al., 2014). Ammonium





($NH_4$) and nitrate ($NO_3$) particles are formed through gas-to-particle reactions involving the gaseous precursors sulphuric acid ($H_2SO_4$), ammonia ($NH_3$) and nitric acid ($HNO_3$). A sulphate-nitrate-ammonium thermodynamic equilibrium seems to be a reasonable assumption for a regional climate model. Nitrate particles can also be formed by heterogeneous uptake of nitric acid over calcite particles ($CaCO_3$, component of dust) and sea-salt ($NaCl$). A&N particles will therefore depend on their precursors but also on sulphates, dusts and sea-salts as described below.

Two size bins for nitrates (fine and coarse modes) and one size bin for ammonium particles were added to the TACTIC aerosol scheme. The chemical formation of A&N particles by gas-to-particles reactions goes into the accumulation mode (fine mode, 0.03 to 0.9 μm). Particles from heterogeneous chemistry correspond to the coarse mode (0.9 to 2.0 μm). For the moment, only the direct effect is taken into account for A&N aerosols, the indirect effect will be taken into account in a future version. The aerosol scheme contains also a new passive tracer for $NH_3$. These new species are subject to the same transport and mixing processes as the other tracers in the ALADIN-Climate model.

It has to be noted that organic nitrates, which might significantly contribute to the fine nitrate aerosol (Kiendler-Scharr et al., 2016) are not included in the model.

### 2.2.1 $HNO_3$ and $NH_3$ auxiliary data

As no full chemistry module is available in ALADIN for reasons of numeric cost, the climatology of $HNO_3$ used in this study is taken from the CAMS Reanalysis (Flemming et al., 2017). A monthly climatology was built over the period 2003-2007 and the annual cycle is taken from nitric acid climatology of Kasper and Puxbaum (1998), based on observations, in order to represent the annual cycle as better as possible. The study of Kasper and Puxbaum (1998) started in November 1991 and covers two annual cycles until November 1993. It is assumed that the nitric acid annual cycle reported in this study is representative of the study area as it was conducted at the Sonnblick Observatory (SBO), which is a high alpine site in the center of Europe, hence relatively distant from pollution sources. To apply the annual cycle of Kasper and Puxbaum (1998) to the $HNO_3$ climatology, we calculated the annual average of $HNO_3$ from CAMS over the period 2003-2007, then we applied to it the new annual cycle, month by month, proportionally to the monthly averages of Kasper and Puxbaum (1998). The $HNO_3$ annual cycle used in this work is presented in Figure 3 (solid line). The $HNO_3$ dataset is assumed constant over the period 1979-2016. Furthermore, there is not a day-night cycle, which can influences the $HNO_3$ content because of a specific tropospheric chemistry during the night (Dimitroulopoulou and Marsh, 1997), in the $HNO_3$ climatology used.

The $NH_3$ used in this work is taken from CMIP6 data and its annual cycle was defined using MACCity emissions dataset. MACCity emissions have been provided in the frame of two projects (MACC and CityZen) funded by the European Commission. To apply the annual cycle of MACCity data to the $NH_3$ emissions dataset, we calculated the annual average of $NH_3$ for each year over the period 1979-2016, then we applied to it the new annual cycle, month by month for each year and at each grid point, proportionally to the monthly averages of MACCity. The $NH_3$ annual cycle used in this work is presented in Figure 2 (red line). Contrary to the $HNO_3$, the $NH_3$ dataset has a year-to-year variability throughout the 1979-2016 period.

Different sensitivity studies (presented in Section 3.2), mostly focused on changes in $NH_3$ emissions (NIT_2, with a different $NH_3$ annual cycle) and $HNO_3$ climatology (NIT_3, with a flat $HNO_3$ annual cycle) will be used in this work.





### 2.2.2 Gas-to-particle reactions

A specific routine presented here aims at equilibrating the gaseous and particle forms of the nitrate and ammonium species. The following equations are based on the study of Mozurkewich (1993). More details about this dependence can also be found in Seinfeld et al. (1998).

Firstly, the $NH_3$ present in the troposphere will react with $H_2SO_4$ to produce ammonium sulphate $((NH_4)HSO_4)$. The reaction for the ammonium sulphate formation will depend on the relative ammonia and sulphate concentrations (Metzger et al., 2002). The total ammonia (TA), total sulphate (TS) and total nitrate (TN) concentrations (kg m$^{-3}$) are defined as:

$$T_A = [NH_3] + [NH_4^+],$$
$$T_S = [SO_4^{2-}],$$
$$T_N = [HNO_3] + [NO_3^-].$$

Ammonium sulphate formation reactions will be different in sulphate-very rich ($T_S > T_A$), sulphate-rich ($T_S < T_A < 2T_S$) and ammonia-rich $T_A > 2T_S$) conditions.

Secondly, if all $NH_3$ is consumed by the ammonium sulphate formation, no A&N aerosols are formed but if there is still some $NH_3$, it will neutralize the $HNO_3$. $K_p$, the equilibrium constant of the reaction of A&N formation is very dependent of relative humidity (RH) and temperature as detailed below. Its formulation is given by the following relations :

$$K_p = \begin{cases} exp[118.87 - 24084/T - 6.025\,ln(T)] & \text{if RH} < \text{DRH} \\ exp[118.87 - 24084/T - 6.025\,ln(T)](p1 - p2RH_1 + p3RH_1^2)RH_1^{1.75} & \text{if RH} \geq \text{DRH} \end{cases}$$

where T is the air temperature (K), DRH the deliquescence relative humidity (DRH, %) calculated by:
DRH = exp (723.7/T + 1.6954)

$RH_1$ = 1-RH/100 and p1, p2 and p3 provided by:
p1 = exp [-135.94+8763/T+19.12 ln(T)],
p2 = exp [-122.65+9969/T+16.22 ln(T)],
p3 = exp [-182.61+13875/T+24.46 ln(T)].

The ammonia available to neutralize the $HNO_3$ is defined as the total ammonia minus the ammonia required to neutralize the available sulphate:

$$T_A^* = T_A - \Gamma\,T_S.$$



With sulphate state $\Gamma = 1$ in sulphate very rich conditions, 1.5 in sulphate rich conditions and 2 in ammonia rich conditions. If $T_N T_A^* > K_p$, the A&N concentration (kg m$^{-3}$) is calculated by:

$$[NH_4NO_3] = \frac{1}{2}\left[T_A^* + T_N - \sqrt{(T_A^* + T_N)^2 - 4(T_N T_A^* - K_p)}\right].$$

If $T_N T_A^* \leq K_p$, A&N dissociate and $[NH_4NO_3] = 0$. The concentration of NH$_3$ (kg.m$^{-3}$) at equilibrium can also be computed with this formula. The NH$_4$ concentration (kg m$^{-3}$) is calculated by:

$$[NH_4] = T_A - [NH_3]$$

The formation of accumulation mode, by gas-to-particles reactions, is faster than the coarse mode because the equilibrium is reached faster by small particles.

This equilibrium leads to the formation of nitrate and ammonium aerosols. Note that this routine is called twice in the TACTIC scheme. Indeed, the first call can be seen as the way to remove the quantity of HNO$_3$ used to form nitrate and

15 ammonium particles in the previous time step, and the second call is the method to form them in the present time step. This is coherent with the fact that the HNO$_3$ comes from the CAMS reanalysis which has no nitrate particles.

### 2.2.3 Heterogeneous chemistry

After the small particles are in equilibrium, the formation of coarse particles by reaction of HNO$_3$ with calcite (part of mineral dust) and sea-salt particles occurs.

A standard first-order reactive uptake parametrization is used to represent the uptake of nitric acid on dust and sea-salt particles and therefore the formation of coarse nitrate particles (Dentener and Crutzen, 1993; Bauer et al., 2004; Fairlie et al., 2010). The loss of HNO$_3$ from the gas phase is represented by the following rate constant k calculated by:

$$k = 4\pi \left(\frac{MMD}{2}\right)^2 N \left(\frac{MMD}{2D_g} + \frac{4}{\vartheta\gamma}\right)^{-1}$$

N is the number density of dust or sea-salt particles of radius [r, r+dr], MMD is the mass median diameter, $D_g$ is the molecular diffusion coefficient that is pressure and temperature dependent. $\vartheta$ corresponds to the mean molecular speed that is also temperature-dependent and $\gamma$ is the reactive uptake coefficient.

One of the limitation of this scheme is the lack of dependence of the uptake coefficient on the aerosol chemical composition

which is not available in ALADIN-Climate. To compensate for that, a Ca$^{2+}$ limitation for the uptake of HNO$_3$ on dust is introduced. Based on dust source maps published by Claquin et al. (1999), we make the assumption that Ca$^{2+}$ represents 5 % of total dust mass. No alkalinity limitation is taken into account for sea-salt particles.



### 2.2.4 Aerosol properties for A&N

The optical properties used as input for SW radiative transfer calculations of A&N particles (0.18-4.0 µm) have been computed using a Mie code (Toon and Ackerman, 1981). As such aerosols are known to be hydrophilic (Tang, 1979), their sizes, density and optical properties are dependant of ambient (grid-box mean) relative humidity (Hauglustaine et al., 2014). In the ALADIN-

Climate model, values of the mass extinction efficiencies, asymmetry parameter and single scattering albedo are tabulated for twelve values of relative humidities: from 0 to 80 % (10 % increments) and from 80 to 100 % (5 % increments). The values of hygroscopic growth factor and optical properties for the fine mode of nitrates are the same as those used in GLOMAP (Manktelow et al., 2010). For coarse mode nitrate, the optical properties are taken from Moffet et al. (2008) and the hygroscopic growth from Gibson et al. (2006). For ammonium aerosols, optical properties used for sulphate are adapted, using a different

molar mass ($NH_4$ against $NH_4(SO_4)^2$ for sulphate). For computations of A&N particles $AOD_{550}$ to be compared with remote sensing data, we use mass extinction efficiencies ($m^2$ $g^{-1}$) as a function of the relative humidity in each model layer. Values for ammonium aerosols are comprised between 4.3 (dry state) and 34.9 (with RH = 95%). For fine (coarse) nitrates, values are comprised between 4.7 (dry state) and 34.2 (with RH = 95%) (between 0.19 and 0.91).

For the fine mode of A&N, a deposition velocity of 0.15 cm $s^{-1}$ is set over all surfaces (ocean, sea-ice, land and land-ice)

for the dry deposition. In comparisons, sulphate dry deposition velocity is set to 0.05 cm $s^{-1}$ over ocean and 0.25 cm $s^{-1}$ over land and land-ice in ALADIN-Climate. The dry deposition for the coarse mode of nitrate is fixed at 1.5 cm $s^{-1}$, which is close to the values used for coarse dust and sea-salt in the model. In ALADIN-Climate, $NH_3$ has the same dry deposition velocity value as sulphur dioxide (between 0.1 cm $s^{-1}$ and 1.5 cm $s^{-1}$ according to the surfaces). The sedimentation is also applied for the coarse mode nitrate (0.13 cm $s^{-1}$). In terms of scavenging, the two bins of nitrate and ammonium have an efficiency for

in-cloud scavenging of 0.8.

## 3 Methodology

### 3.1 Observations

In this study, different datasets have been used to evaluate the ability of the ALADIN-Climate model at reproducing the A&N concentrations at the surface as well as the total AOD, before investigating the radiative and climatic impacts in present-day

conditions. In that context, two monthly satellite datasets have been used to provide a regional estimate of the total $AOD_{550}$.

Firstly, the MODerate resolution Imaging Spectroradiometer (MODIS, collection 6.1, 1°resolution; (Tanré et al., 1997; Sayer et al., 2014)) is used. MODIS is a 36-band polar orbiting radiometer aboard both EOS Aqua and Terra (both separately used in this work), with equatorial crossing times of about 10:30 and 13:30, respectively. The MODIS aerosol products are generated from different well-known algorithms, including the dark target (DT) algorithms over both the oceans and land and the deep

blue (DB) algorithm over only land (Kaufman et al., 1997). They have an uncertainty on AOD at 550 nm of approximately ± 0.03 (Sayer et al., 2013). MODIS Terra covers the 2001-2016 period and MODIS Aqua the 2003-2016 period.





Secondly, the Multiangle Imaging SpectroRadiometer (MISR, Level 3, $1°$ resolution; (Kahn and Gaitley, 2015)) has been used to provide a regional estimate of the total $AOD_{550}$. MISR is a polar orbiting instrument aboard EOS Terra. The MISR aerosol product provides aerosol distributions over both land and oceans. The MISR dayside equator crossing is at about 10:30 AM local time. It has an uncertainty in AOD at 550 nm of approximately $\pm$ 0.05 (Kahn et al., 2010). MISR dataset covers the

2001-2016 period. Both satellites (MODIS and MISR) are available at the NASA Earthdata portal.

In parallel, the measurements obtained from the AErosol RObotic NETwork (AERONET) network provide local column-integrated aerosol properties like the total AOD at different wavelengths (Holben et al., 2001). Six stations (Figure 1, black triangles) with a long series of data (at least 5 years) were chosen to best represent the domain used: three in the north of Europe in Cabauw, Hamburg and Belsk (Netherlands, Germany and Poland), one in the south in Barcelona (Spain), one in

the east in Sevastopol (Crimea) and one in North Africa in Blida (Algeria). These stations are detailed in Table 1. These sun-photometer observations provide high-quality data. Version 2, Level 2 AOD have been downloaded from the AERONET website (https://aeronet.gsfc.nasa.gov). For comparison to our model results, all AOD data have been calculated at 550 nm using the Ångstrom coefficient between the closest available upper and lower wavelengths and we made monthly averaging of original AERONET surface observations. The AOD uncertainty in version 2 (Level 2) AERONET data is $\pm$ 0.01 in the visible

(Eck et al., 1999).

Satellite and AERONET AOD data are obtained during daytime only (even at a given hour for satellites). On the other hand, our AOD averages from our simulations were obtained on the whole day (night plus day), which is therefore a source of uncertainty to take into account.

Concerning the evaluation of the A&N surface concentration, the ground-based station network EMEP (The European

Monitoring and Evaluation Programme), using standardized monitoring method and analytical techniques over Europe, has been used (Tørseth et al., 2012). Such surface in-situ observations are very useful to evaluate regional climate model as the EMEP stations are located in remote areas representing a larger region, avoiding influences and contamination from local sources (Bian et al., 2017). Contrary to AOD, which is related to the total aerosol column load, they enable us to evaluate the A&N aerosols only with the surface concentration of the nitrate and ammonium aerosols (total suspended particulate). These

stations do not have continuous data over the period 1994-2014 and we selected those with a minimum of 5 years of data, which may be non-continuous but a minimum of 5 observations is necessary for every month. Finally, 33 stations were selected for nitrate and 35 for ammonium, symbolized by black dots in Figure 1. These stations are detailed in Table 2. For comparison to our model results, we made monthly averaging of original EMEP surface observations.

### 3.2   Simulations

Two main configurations have been used for the ALADIN-Climate simulations in the present work, including or not A&N aerosols. The simulation defined as the reference for this study is called REF. It has used the ALADIN-climate model described previously including all aerosols except A&N. The second simulation, called NIT, is the same simulation but including the new A&N aerosol module. Both of them cover the same period from 1979 to 2016. These two simulations are driven by the ERA-Interim reanalysis both for the lateral boundary conditions and inside the domain using the spectral nudging method described





in Radu et al. (2008). This method allows us to better impose the large scales from the boundary forcing dataset and therefore better follow the true natural climate variability. The wind vorticity and divergence, the surface pressure, the temperature and the specific humidity are nudged. A constant rate above 700 hPa and a decreasing rate between 700 and 850 hPa are imposed, while the levels below 850 hPa are free. The spatial wavelengths are similarly nudged beyond 400 km, with a transition zone

between 200 and 400 km. Finally, this method gives the model enough freedom to generate the aerosols at the surface while keeping the ERA-Interim large scale conditions that are required to simulate the true chronology.

Additional simulations have been performed for different sensitivity studies, mostly focused on changes in $NH_3$ emissions and $HNO_3$ climatology. Firstly, the impact of $NH_3$ emissions has been investigated using the NIT_2 simulation, which covers the 1979-2016 period. This additional run is similar to the NIT simulation and differs only on the annual cycle used for $NH_3$

emissions. In that sense, NIT_2 uses $NH_3$ emissions which are taken from CMIP6 data (as the NIT simulation) but without the annual cycle of $NH_3$ MACCity emissions. The annual cycle of $NH_3$ emissions used for both simulations is shown in Figure 2. $NH_3$ emissions of the NIT simulation (MACCity annual cycle) present an earlier maxima (March) than the one observed for the NIT_2 simulation (CMIP6 raw data) which have a maximum in May. In a second time, we intend to estimate the impact of the seasonality of the $HNO_3$ climatology. Indeed, the $HNO_3$ of the NIT simulation presents a relatively strong peak in April

and a second weaker peak in July-August (Figure 3). In order to evaluate the impact of this annual cycle, the simulation NIT_3 has been realized using a flat annual cycle of $HNO_3$. All the ALADIN-Climate simulations are summarized in Table 3.

## 4  Evaluation of the new A&N aerosol scheme

### 4.1  Surface concentration

The A&N aerosol concentrations of the NIT simulation, simulated at the surface by ALADIN-Climate, are evaluated in this

section against observations obtained at EMEP stations over the period 1994-2014. Figure 4 presents comparisons of A&N concentrations simulated by the model with the concentrations measured at EMEP stations. The nitrate concentration simulated by ALADIN-Climate corresponds to the cumulated concentration of the accumulation and the coarse mode.

First, it should be mentioned that the model is able to reproduce some areas with high concentrations of nitrate particles (Benelux and Po Valley), with values of about 6 µg m$^{-3}$ for the Benelux and 10 µg m$^{-3}$ for the Po valley. Then, the spatial

correlation calculated between the model and the EMEP stations is found to be about 0.82 for nitrate concentration. The patterns of the simulated surface nitrate concentrations are in general agreement with different global model results (Myhre et al., 2006; Bauer et al., 2007) and also with the chemistry-transport model LOTOS (Schaap et al., 2004) that show annual average nitrate concentration (year 1995) between 5 and 8 µg m$^{-3}$ over the Benelux. Schaap et al. (2004) also report elevated concentrations over the Po valley, where the annual averaged concentrations exceed 4 µg m$^{-3}$. Furthermore, Hauglustaine

et al. (2014) showed maximum concentrations in the same order of magnitude as the ALADIN-Climate model around 4-5 µg m$^{-3}$ over northern Europe. Figure 2 also indicates that the concentrations calculated over regions near the source areas (Benelux and Po Valley) are generally overestimated by the model, especially in Italy and eastern Europe. For example, the Montelibretti station (near Rome) is characterized by nitrate concentrations of about 4 µg m$^{-3}$ while the model simulates a




surface concentration of about 8 µg m$^{-3}$. Other stations, particularly in Italy, would be needed to confirm this overestimation by ALADIN-Climate. Furthermore, outside of Europe and more specifically over the western Mediterranean where surface A&N concentrations simulated by ALADIN-Climate are relatively high, there are no available stations for evaluating the simulations. These high concentrations might be due to strong concentrations of nitric acid in the CAMS climatology.

Concerning the ammonium concentration, the model is found to be generally close to in-situ EMEP observations with a spatial correlation of 0.86. However, underestimates are detected over northern Europe and Benelux. The regional pattern of the surface ammonium concentrations is found to be in agreement with results shown by Hauglustaine et al. (2014) (2-3 µg m$^{-3}$ in northern Europe) and also with results presented by Schaap et al. (2004), who show annual average ammonium concentration (for the year 1995) around 2-3 µg m$^{-3}$ over the Po valley.

To summarize, the analyses of comparisons with EMEP dataset demonstrate the ability of the model to reasonably reproduce the ammonium concentrations at the surface over the European/Mediterranean region, notably their spatial distribution. However, some regions such as Eastern Europe or Italy are concerned with a positive bias in nitrate concentrations.

## 4.2   AOD at local scale (AERONET)

In addition to surface concentrations, comparisons have also been realized at local scale using different AERONET stations.
These stations, with at least five years of data, were chosen to cover the domain used and include different contrasted aerosol regimes. These stations are represented in Figure 1 by black triangles. Figure 5 reports comparisons of the average annual cycle of the total AOD measured at AERONET stations with the one simulated by ALADIN-Climate, including the contribution of each aerosol type. AOD data presented here have been calculated at 550 nm over the period 2003-2012. It is important to note that the total nitrate AOD is primarily influenced by the first nitrate bin, which has much higher extinction per mass
(5-30 m$^2$ g$^{-1}$ at 550 nm depending on RH) than the second bin (0.20-0.90 m$^2$ g$^{-1}$). Previous studies have already shown the predominance of nitrates in the fine fraction of aerosols (Schaap et al., 2002). A first important point is that the A&N contribution to the total AOD$_{550}$ is found to be significant, especially near source areas (Benelux, Po valley) as Cabauw, Belsk and the Hamburg stations. For example, for Cabauw, Belsk or Hamburg, A&N represents more than half of the total AOD$_{550}$ during spring and summer with values comprised between 0.10 and 0.25. The maxima in A&N AOD$_{550}$ is generally observed
during spring (March, April, May) with values reaching 0.25 at the Cabauw or Hamburg AERONET stations. Indeed, the seasonal cycle of A&N AOD$_{550}$ follows the same seasonal cycle of NH$_3$ and HNO$_3$ emissions, which are characterized by highest values during spring. In parallel, stations far from the emission sources (Blida, Sevastopol) are characterized by low A&N AOD$_{550}$ of 0.02 throughout the year, with a maximum in March (0.10). For all the stations, the minimum of A&N AOD$_{550}$ is obtained in winter and the maximum in spring as reported in the Table 4 which presents the average seasonal values
of the A&N AOD$_{550}$ simulated by the model (NIT simulation) at three stations, Cabauw (northern Europe), Barcelona and Sevastopol in eastern Europe (far from the sources). Table 4 indicates clearly that the highest values occur in spring with 0.17 for Cabauw (close from the sources) and 0.05 for Sevastopol (far from the sources). The minimum values are obtained in winter, with 0.05 for Cabauw and 0.01 for Sevastopol. Barcelona is characterized by intermediate values of 0.15 in spring and 0.07 in winter.



In the majority of cases, as shown in Figure 5, the ALADIN-Climate simulations demonstrate that the contribution of A&N $AOD_{550}$ improves the average AOD annual cycle, as for Sevastopol, Cabauw or Barcelona AERONET stations. This result highlights that A&N aerosols are important to take into account for correctly simulating the spatial and temporal variability of the total $AOD_{550}$ over the Euro/Mediterranean region. Nevertheless, some bias are identified and the total $AOD_{550}$ simulated

by the ALADIN-Climate model is sometimes too weak in summer as for the Cabauw station. This lack of aerosols in summer can be also explained by the absence of anthropogenic secondary organics aerosols (SOA) in the ALADIN-Climate model, since only the natural SOA are considered through the climatology of Dentener et al. (2006). It can be also explained by the absence of organic nitrates, which might significantly contribute to the fine nitrate aerosol (34% to 44% of submicron aerosol nitrate) in regions with high nitrate concentrations (Kiendler-Scharr et al., 2016). Furthermore, such local comparisons between

the model (with an horizontal resolution of 50 km) and AERONET data are related to uncertainties due to the representative-ness of sun-photometer observations when compared to the model grid point mean.

## 4.3  AOD at the regional scale

In addition to local comparisons at EMEP and AERONET stations, we have extended our analyses to the regional AOD

spatial distribution using different satellite products (MODIS and MISR). The average total $AOD_{550}$ for the REF (without A&N particles), the NIT simulation (including A&N) and the satellite observations over the period 2001-2016 (2003-2016 for MODIS Aqua) are summarized in Figure 6.

First, Figure 4 indicates that the regional spatial pattern of $AOD_{550}$ simulated by the ALADIN-Climate model is improved over Europe in the NIT simulation relatively to the REF simulation when compared to satellite data (MODIS and MISR).

As A&N aerosol concentrations are found to be less important over the Mediterranean Sea and Africa, there are logically few differences between the NIT and REF simulations in these regions. Over Europe, Figure 7 shows that the additional $AOD_{550}$ due to A&N aerosols can explain part of the negative bias in the REF simulation, especially concerning the Benelux and the Po Valley. The REF simulation presents $AOD_{550}$ equal to 0.09 (at 550 nm) in average over Europe while the NIT simulation presents higher values (0.16 at 550 nm) in a better agreement with satellite data ranging from 0.13 to 0.19. It should

be noted however that the different satellite data show large differences between themselves. Concerning the total $AOD_{550}$ over the Mediterranean sea, satellite data (MODIS and MISR) indicate values between 0.20 and 0.22. In this case, the NIT simulation shows a mean value of 0.22, producing a slight improvement compared to the REF simulation (mean of 0.19). All the results are summarized in Table 5 for the different ALADIN-Climate simulation and satellite data. Both domains (Europe and Mediterranean Sea) are represented in Figure 1. Compared to other modelling studies including A&N aerosols over the

Euro/Mediterranean region, the simulated A&N $AOD_{550}$ by the ALADIN-Climate model are found to be consistent with those reported by Hauglustaine et al. (2014) with values comprised between 0.15 and 0.25 over Europe. More specifically, Figure 7 indicates highest values of A&N $AOD_{550}$ over the Benelux (0.07) and the Po valley (0.09) which are a little bit higher than Hauglustaine et al. (2014), who report a maximum about 0.05 in northern Europe. Figure 7 also shows significant values over the Red Sea (near 0.1), which are certainly due to high $HNO_3$ concentration over this region. It should be noted that Ammonium



AOD$_{550}$ presents the same spatial distribution as nitrates over Europe, with maxima over the Benelux and the Po valley but lower in magnitude than nitrates. Indeed, Ammonium AOD$_{550}$ reach up to 0.06 over the Po Valley and 0.04 over the Benelux.

## 4.4   AOD annual cycle at regional scale

The annual cycle of the total aerosol AOD$_{550}$ (2001-2016) simulated by ALADIN-Climate and measured by satellite instruments (MODIS and MISR) over the two different domains considered (Europe and the Mediterranean Sea; see Figure 1) is presented in Figure 8. The model results indicate that the A&N AOD$_{550}$ represent 40 % of the total AOD over Europe with an average of 0.07, demonstrating the importance of A&N particles in terms of radiative budget over this region. Unlike to REF, A&N AOD$_{550}$ over Europe in the NIT simulation is found to be in the same order of magnitude as satellite observations. Over Europe, the NIT simulation in Figure 6 also indicates that A&N aerosols improve significantly the simulated total AOD$_{550}$ annual cycle compared to satellite observations. Indeed, the increase in AOD$_{550}$ from winter to spring is now more important in NIT (0.11 to 0.21) than in REF (0.08 to 0.10), which is in better agreement with the different satellite datasets, ranging from 0.08-0.11 in winter to 0.15-0.21 in spring. Moreover, the NIT simulation presents two peaks in April and in August as in MISR data. However, the annual cycle of MODIS (Terra/Aqua) is found to be significantly different from MISR, with a single maximum during the summer period. The NIT simulation also presents a positive bias in spring (February, Mars and April) and a negative bias in summer (May, June and July) compared to MODIS data. Over the Mediterranean Sea, AOD$_{550}$ is overestimated during spring (Mars, April and May) both in REF and NIT simulations, and underestimated during summer, probably due to discrepancies in the annual cycle of dust aerosols. This overestimate during spring and underestimate during summer can be seen in Figure 5 with the Blida station located in northern Algeria near the Mediterranean Sea.

Despite this bias, the peak present in MODIS datasets in April is well reproduced by ALADIN, and highlighted in the NIT simulation. Nevertheless there is no clear improvement in AOD$_{550}$ over the Mediterranean Sea when adding nitrate aerosols.

## 4.5   Sensitivity tests

The different biases discussed previously can be due to numerous uncertainties, such as the dry or wet deposition, the hygroscopic and optical properties, and especially the precursor species of A&N aerosols (NH$_3$ emissions or HNO$_3$ concentrations). For this latest reason, two sensitivity tests were carried out over Europe. The first concerns the annual cycle of NH$_3$ emissions and the second the annual cycle of HNO$_3$.

Figure 9 shows the impact of the change in NH$_3$ emissions (NIT_2 simulation) and HNO$_3$ concentration (NIT_3 simulation) at four EMEP stations. These stations, with continuous data over 16 years, are located in the Netherlands, Italy, Turkey and Russia to represent different aerosol regimes. At these stations, the nitrate concentration estimated from the EMEP network is fairly stable during the year with 6 µg m$^{-3}$ at the beginning of the year and about 3-4 µg m$^{-3}$ at the end at De Zilk where the amplitude is maximum. In the eastern part of the domain (Cubuk and Danki stations), the nitrate concentrations observed are very low and comprised between 0 and 1 µg m$^{-3}$ throughout the year. The results indicate that the NIT simulation, as well as the two sensitivity tests (NIT_2 and NIT_3), are quite different from the observations at these four stations. Indeed, the ALADIN-Climate model presents an overestimate of the nitrate surface concentration especially with a high peak during



the spring (March, April, May). Except at the De Zilk station, the NIT simulation presents an earlier peak in the spring than NIT_2 simulation due to its annual cycle of NH_3 emissions (Figure 2). At the De Zilk station, the NIT_3 simulation with a flat annual cycle of $HNO_3$, shows nitrate concentrations at the surface closer to EMEP observations, with relatively constant values ranging from 5 to 8 µg m$^{-3}$. For other stations, this simulation reveal overestimates of nitrate surface concentrations

compared to EMEP data. Several parameters may explain this overestimate of the simulated nitrate concentration as excessive emissions of ammonia or too high nitric acid concentration in the climatology used in the ALADIN-Climate model. Indeed, Bian et al. (2017) have shown that a good nitrate simulation depends on good simulations of precursors, such as $NH_3$ and $HNO_3$.

Concerning the AOD, Figure 10 presents the annual cycle of total $AOD_{550}$ for NIT, NIT_2 and NIT_3 simulations compared

at four AERONET sites with continuous data over 10 years (Netherlands, Spain, Crimea, Poland). Differences between NIT, NIT_2 and NIT_3 simulations are therefore due to A&N $AOD_{550}$ differences. Contrary to surface nitrate concentrations, the A&N $AOD_{550}$ is found to be closed to AERONET observations, especially for the NIT simulation at the Cabauw or at the Belsk stations. Moreover, at Barcelona, the model shows an overestimate during spring (peak of 0.29 for NIT and NIT_3 and in May (maxima of 0.36 for NIT_2 simulation). For these periods, AERONET data indicates AOD of about 0.16.

In addition, Figure 8 presents the total aerosol $AOD_{550}$ simulated by NIT (red), NIT_2 (blue) and NIT_3 (green) simulations compared to satellite products (grey dots) over Europe (2001-2016). The three simulations are found to be close but NIT_2 simulation presents a higher and later peak of around 0.26 in May than the NIT simulation, which reaches maximum (0.24) in March-April over the Europe domain. During the May to July period, the NIT simulation shows lower values. The NIT_3 simulation shows a weaker peak in April (0.20) than the NIT simulation (0.24). For the rest of the year, NIT and NIT_3

simulations are very close. Table 6 presents the temporal correlation between mean annual cycles from different simulations (NIT, NIT_2 and NIT_3) and satellite products (MISR, MODIS Aqua and MODIS Terra). Table 6 indicates that NIT simulation, with a peak in April, is closer to MISR than NIT_2 simulation. NIT and NIT_2 simulations present a positive bias (0.04) compared to MISR dataset. In parallel, the NIT_2 simulation is found to be closer to MODIS Aqua and MODIS Terra than the NIT simulation. These simulations present also a positive bias (0.02 for NIT and 0.01 for NIT_2) compared to MODIS Aqua

and a negative bias compared to MODIS Terra (- 0.02). The annual cycle of the $NH_3$ emissions has therefore a significant impact on the total aerosol $AOD_{550}$ especially during the spring (March, April and May). Concerning NIT_3, as shown in the table 6, this simulation presents high temporal correlation compared to MISR (0.95), associated to a bias of 0.03. This run shows also better correlations with MODIS Aqua (0.82) and MODIS Terra (0.85), than the NIT simulation (0.75 and 0.77 respectively).

Finally, it is shown here that the annual cycles of nitrate precursors, such as ammonia and nitric acid, have significant impacts on the A&N $AOD_{550}$ and therefore on the total AOD, especially in spring. Hence, a poor representation of the annual cycle of nitrate precursors may therefore be one of the possible causes of the AOD bias observed in spring compared to satellite data (Figure 8). Further studies related to the representation of the ammonia emissions and nitric acid annual cycle seem necessary to improve the simulated A&N concentrations and optical depth over Europe.



## 4.6 Aerosol trends

Figure 11 presents the total $AOD_{550}$ evolution over Europe and Mediterranean sea for REF and NIT simulations. NAB2013 is an aerosol climatology developed by Nabat et al. (2013), which is based on MODIS data for total AOD over the period 2003-2009, and model data for the distinction in the contribution of the different aerosol types. Before 2003, this climatology

is extended up to 1979 using the sulphate AOD trend coming from the LMDz-INCA model (which do not have AN aerosols), in agreement with the other ACCMIP models. Figure 11 indicates an important decrease over Europe of the total $AOD_{550}$ from 1979 to 2016 in the REF (- 0.047 per decade) and NIT (- 0.035 per decade) simulations and also from NAB2013 (- 0.045 per decade). Figure 11 shows also a less pronounced decrease of the total $AOD_{550}$ over the Mediterranean sea. Such decreases are mainly due to the decline of sulphate particles over Europe during this period. The different trends obtained for

the NIT simulation (Total, sulphate and A&N trends) and NAB2013_sulphate (sulphate trend) for the period 1979-2016 are summarized in Table 7. First, NAB2013_sulphate that does not take into account A&N particles, is very close to the sulphate trend of NIT simulation over Europe (-0.046 for NIT and -0.045 for NAB2013) and Mediterranean (-0.022 for NIT and -0.021 for NAB2013). On the other hand, over the period 1979-2016, the NAB2013_sulphate trend is found to be about 30% stronger than the NIT total run over Europe and about 50% stronger over the Mediterranean sea, notably because of the positive A&N

trend of 0.012 per decade over Europe (and 0.008 per decade over the Mediterranean sea). For comparisons, the nitrate trend obtained from the GISS global model (Shindell et al., 2013) for the period 1980-2015 is found to be relatively similar to the NIT simulation, with lower values like 0.008 per decade over Europe (0.012 for the nitrate trend of the NIT simulation) and 0.005 per decade over the Mediterranean (0.008 for the nitrate trend of the NIT simulation).

In Figure 11, satellite data (MODIS and MISR), highlighted by the shaded area, also indicate a decrease in total AOD

between 2003 and 2015, when averaged over Europe. Trends obtained for the different simulations and from satellite products, over the period 2003-2015, are also presented in Table 8. This decrease in total aerosol AOD is also due to the strong drop of sulphate aerosols. Over Europe, unlike REF simulation, NIT simulation is of the same order of magnitude as the satellite products. On the other hand, it is shown that over Europe but also over the Mediterranean sea, the trend of the NIT simulation is weaker than the REF simulation and all satellite data. The difference between NIT and REF is due to the positive trend

of nitrate and ammonium (0.013 per decade over Europe and 0.011 per decade over the Mediterranean sea). Differences between NIT and satellite products may be due to $NH_3$ emission inventories or due to $HNO_3$ climatology (uncertainties in $NH_3$ emission inventories and $HNO_3$ annual cycle averaged over 5 years). Furthermore, the AOD quantification by satellite products is difficult because the aerosol contribution to the reflectance is mixed with that of clouds and the surface (Bréon et al., 2011) and also because they do not take into account all the pixels (especially with MISR).

The lower trend obtained for the NIT simulation compared to the REF run in Figure 11 is therefore due to the continuous increase of nitrate aerosol concentrations between 1979 and 2016, from 0.025 (1979) to 0.06 (2016) over Europe and from 0.01 (1979) to 0.03 (2016) over the Mediterranean sea. This increase in nitrate concentrations then partially compensates the sulphate concentration decrease. An interesting point is that the nitrate $AOD_{550}$ rise is not due to an increase in its precursors (ammonia and nitric acid). Indeed, the nitric acid used in this study is constant over the years and secondly because, as shown





in Figure 12, ammonia emissions are also constant or even slightly lower since 1979 over Europe. Figure 12 also shows that the decrease in sulphate aerosols observed over Europe is due to the decline of one of its precursors (sulphur dioxide). This decrease in sulphate aerosol production thus leaves more free ammonia in the lower atmosphere allowing an increase in A&N aerosols over Europe but also over the Mediterranean. Figure 11 shows the relative importance of the A&N $AOD_{550}$ over

5 Europe, which is higher than sulphate AOD since 2005 in our NIT simulation. To our knowledge, this is the first report that A&N aerosols appear as the most important species in terms of $AOD_{550}$ over Europe from 2005 on.

## 5  Impact of ammonium and nitrate aerosols on the radiative budget and regional climate

### 5.1  Direct SW radiative forcing

This final section aims at analysing the impact of A&N aerosols on the European/Mediterranean radiative budget and its

climate, notably in terms of surface temperature. First, Figure 13 presents the impact of A&N aerosols on the solar radiation at the surface and at the TOA for the period 1979-2016 for clear-sky and all-sky conditions. Figure 13 indicates a moderate effect of A&N aerosols on the surface SW radiation over central Europe, of about -4 W m$^{-2}$ in clear sky and -2 W m$^{-2}$ in all sky conditions, respectively. This Direct Radiative Forcing (DRF) is consistent with the A&N $AOD_{550}$ (0.1-0.15) simulated over this region. The most marked season is summer (June, July and August), shown at the top of Figure 15, with values up to -5 W

m$^{-2}$ over central Europe and -10 W m$^{-2}$ in the Po Valley at the surface in all sky conditions.

When averaged over Europe, A&N aerosols cause a surface DRF of around -1.7 W m$^{-2}$ (all-sky conditions). Figure 13 also reveals local maxima in the surface forcing (-5 W m$^{-2}$) over Europe (all-sky conditions) especially in the Po Valley and the Benelux, certainly because of the presence of industrial pollution particles. Over the Po Valley, the A&N surface DRF represents 33% (all-sky conditions) of the total surface DRF. It should be finally noted that the maxima of DRF exerted by

20 A&N aerosols is found to occur during summer.

Figure 13 also shows that A&N aerosols cause a mean TOA DRF over Europe of around -1.4 W m$^{-2}$. Bauer et al. (2007) and Hauglustaine et al. (2014) indicate also a significant nitrate TOA DRF over Europe (-1 W m$^{-2}$) consistent with the one estimated in this study. This forcing over Europe is one order of magnitude larger than the global average for present conditions (-0.11 W m$^{-2}$; Bauer et al. 2007). At the TOA, local maxima (-5 W m$^{-2}$) are detected over the Po Valley and

25 Figure 13 indicates important TOA DRF of around -2 W m$^{-2}$ over the Benelux (all-sky conditions).

Table 9 summarizes the calculated annual-mean SW DRF exerted at the surface and at TOA for the REF and NIT simulations over the period 2000-2009, compared with those obtained by Nabat et al. (2015b) CNRM-RCSM4 simulations (period 2000-2009) and by Papadimas et al. (2012) using MODIS (period 2000-2007).

First, it should be mentioned that differences between REF and CNRM-RCSM4 simulations are mainly due to different

versions of the atmospheric model (ALADIN-Climate). In addition, the CNRM-RCSM4 model did not use an interactive aerosol scheme but the Nabat et al. (2013) aerosol climatology. Table 9 reveals also that the NIT simulation improves the DRF estimates, compared to MODIS data. The aerosol DRF evolution over Europe and the Mediterranean sea is presented in Figure 14 in all sky and clear sky conditions, at the surface and at the TOA, over the period 1979-2016. Despite an increase in the



A&N DRF, the total surface DRF is found to decrease over Europe from -12 W m$^{-2}$ in 1979 to -8 W m$^{-2}$ in 2016 in clear-sky conditions and from -8 W m$^{-2}$ (1979) to -5 W m$^{-2}$ (2016) in all-sky conditions. This trend is mainly due to the decrease in sulphate and organic DRF, which is not fully offset by the increase in A&N DRF (from -1.5 W m$^{-2}$ in 1979 to -4 W m$^{-2}$ in 2016 over Europe in clear-sky conditions). An important result here concerns the relative importance of the A&N DRF exerted

over Europe, both at the surface and at TOA, which is found to be higher than sulphate and organic DRF since 2005. Hence, our simulations indicate for the first time to our knowledge, that since this specific year (2005), A&N aerosols appear as the most important species in terms of DRF over Europe. Paulot et al. (2018) have estimated the DRF of these different aerosol from 2001 to 2015 using the GFDL chemistry-climate model AM3 driven by CMIP6 historical emissions. They have also shown a decrease in total aerosol DRF over western Europe driven by the decrease in sulphate associated with the decrease

in sulphur dioxide emissions. On the other hand, they didn't show an increase in A&N aerosols over Europe. In their study, sulphate aerosols always have a stronger AOD and a stronger DRF than nitrate aerosols. In parallel, Figure 14 shows a moderate decrease in the total aerosol DRF over the Mediterranean (both at the surface and TOA), mainly due to the decrease in sulphate and organic DRF. The A&N DRF is also found to increase in this area from -1 W m$^{-2}$ in 1979 to -2 W m$^{-2}$ in 2016, with an equivalent DRF to sulphate and organic aerosols in 2016.

**5.2 Effects on the regional climate**

We investigate here the consequences of the direct radiative forcing of A&N aerosols on near-surface air temperature at 2m and precipitation over the model domain.

Figure 15 presents the differences (averaged over the period 1979-2016) between the NIT and REF simulations for 2m-temperature and surface DRF SW (all-sky conditions) during summer (June, July and August). Figure 15 only indicates signif-

20 icant values at the level 0.05. This figure shows a moderate temperature impact due to A&N surface dimming of about -0.2°C over Europe, due to the radiation drop (-1.7 W m$^{-2}$ on average over Europe). A larger decline takes place (-0.4°C) in regions with large A&N AOD$_{550}$ associated to a significant DRF (Benelux, Po Valley). For other seasons, no significant drop in temperature has been found. As a comparison, Nabat et al. (2015b) reported that 2m-temperature could be reduced by 0.4°C on average over Europe (-0.2°C in winter and -0.4°C for each other seasons) because of the presence of different aerosols (sea-salt,

desert dust, sulphate, black and organic carbon) over the period 2003-2009. Therefore, only for the summer, the A&N impact on 2m-temperature is about half as strong as the effect of all other aerosols. Zanis (2009) has also shown a negative surface radiative forcing associated with the anthropogenic aerosols (carbonaceous particles and sulphates), during the summer 2000 throughout the European domain. Anthropogenic aerosols cause a temperature decrease at the lower troposphere up to 1.2°C over Southeastern Europe and the Balkan Peninsula, about three times more than summer A&N impact on 2m-temperature.

Finally, no significant difference has been found regarding precipitation between REF and NIT simulations (not shown).





## 6 Conclusions

In this work, we have developed a new configuration of the aerosol scheme TACTIC in ALADIN-Climate model, notably by adding ammonium and nitrate (A&N) aerosols. The objective is to present the implementation and the evaluation of this simplified A&N module and to assess the direct radiative effect and climatic impact of A&N aerosols over the Euro-Mediterranean region for past-present conditions. Two main parallel simulations have been realized; the REF simulation which includes all aerosols except A&N (dust, sea-salt, black carbon, organic carbon and sulphate), and the NIT simulation also including A&N particles. Our results indicate that A&N surface concentrations are relatively well represented (maxima over the Benelux and the Po Valley) but some important differences are detected with EMEP data (overestimates over Italy and Eastern Europe notably). For total $AOD_{550}$, A&N aerosols are shown to reduce the model low bias particularly over Europe, notably with the improvement of the annual cycle of total aerosol $AOD_{550}$ (especially in some areas like the Benelux). However, some biases have been identified, as an overestimate during spring of total $AOD_{550}$ over Europe and the Mediterranean and an underestimate during summer over the Mediterranean. Sensitivity studies suggest that such biases are related to uncertainties associated with the annual cycle of A&N aerosol precursors (ammonia and nitric acid).A lack of other aerosols such as secondary organic aerosols or organic nitrates is also possible in the model. Over Europe, the ALADIN-Climate simulation indicates that A&N aerosols are characterized by a positive trend over the period 1979-2015 (0.012) due to the increase in A&N aerosol production. This could be caused by more "free" ammonia in the lower troposphere due to the decrease in sulphate aerosols over the 38 years. The addition of A&N aerosols has allowed us to estimate the impact of A&N aerosols on SW radiations and surface temperatures. At the surface and for all-sky conditions, A&N particles are found to represent about 26 % of the total aerosol DRF over Europe, yielding a decrease in surface. The presence of A&N aerosols is shown to cause a decrease in surface SW radiations of 5 W m$^{-2}$ in all sky conditions in the Po Valley over the period 1979-2016. The analysis of the different aerosol trends over the period 1979-2016 indicates that since 2005, the DRF due to A&N particles over Europe has become more important than those exerted by sulphate and organic particles. Finally, our model results indicate that the impact of A&N aerosol on surface SW radiations causes a cooling of -0.2 degrees during summer (June, July, August). But no significant effect of the DRF of A&N aerosols was found on precipitation. To go further, it would now be interesting to study the role of A&N aerosols over the Euro-Mediterranean region in future climate regional projections, as they are expected to have a larger contribution to anthropogenic AOD by the end of this century (Hauglustaine et al., 2014).

*Data availability.* This study relies entirely on publicly available data. MISR and MODIS AOD products can be obtained from the NASA Earthdata portal. AERONET data are available at AERONET: https://aeronet.gsfc.nasa.gov/ (last access: 28 November 2018). EMEP data are available at EMEP: http://ebas.nilu.no/ (last access: 28 November 2018). Model outputs are available upon request from the authors. The $HNO_3$ comes from the CAMS Reanalysis (Flemming et al., 2017) and its annual cycle comes from Kasper and Puxbaum (1998). $NH_3$ data come from CMIP6 data at: https://esgf-node.llnl.gov/search/cmip6/ and its annual cycle was defined using MACCity emissions dataset at: http://eccad.aeris-data.fr/#DatasetPlace:.



*Author contributions.* All authors designed the simulations and TD carried them out. All authors provided input on data analysis shown in the paper. PN and TD developed the model code. TD prepared the manuscript with contributions from all co-authors.

*Competing interests.* The authors declare that they have no conflict of interest.

*Acknowledgements.* We would like to thank Meteo-France and the Occitania region for the financial support of the first author. This work

5   is part of the Med-CORDEX initiative (www.medcordex.eu) and a contribution to the MISTRALS/ChArMEx programme. We thank the principal investigators of the AERONET and EMEP networks and their staff for establishing and maintaining the different sites used in this investigation. NASA Atmosphere Archive and Distribution System (LAADS, http://ladsweb.nascom.nasa.gov) is acknowledged for making available the MODIS/Terra and Aqua Collection 6.1 aerosol datasets, as well as the MISR/Terra dataset. We also acknowledge the Copernicus Climate Change and Atmosphere Monitoring Services for providing us with the CAMS reanalysis and Samuel Remy for his help

10  in integrating the nitrate and ammonium module in our model.




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

| Station | Location | Altitude (m) | Number of months available over the observation period (2003-2012) |
|---|---|---|---|
| Cabauw (Netherlands) | 51.971N, 4.927E | -1 | 89 |
| Hamburg (Germany) | 53.568N, 9.973E | 120 | 77 |
| Barcelona (Spain) | 41.389N, 2.112E | 125 | 88 |
| Sevastopol (Crimea) | 44.616N, 33.517E | 80 | 82 |
| Blida (Algeria) | 36.508N, 2.881E | 230 | 85 |
| Belsk (Poland) | 51.837N, 20.792E | 190 | 105 |

**Table 1.** Station name, location, altitude and number of months available over the observation period (2003-2012) of the 6 AERONET stations.





| | Station | Location | Altitude (m) | Number of months available over the observation period (2003-2012) | |
|---|---|---|---|---|---|
| | | | | Nitrate | Ammonium |
| Austria | Illmitz | 47.460N, 16.460E | 117 | 97 | 97 |
| Czech Republic | Svratouch | 49.440N, 16.300E | 737 | 84 | 84 |
| | Kosetice | 49.350N, 15.500E | 534 | 84 | 84 |
| Denmark | Tange | 56.210N, 9.360E | 13 | X | 72 |
| | Anholt | 56.430N, 11.310E | 40 | X | 72 |
| | Ulborg | 56.170N, 8.260E | 10 | X | 72 |
| Great Britain | Cough Navar | 54.263N, 7.521W | 126 | 108 | 108 |
| | Yarner Wood | 50.354N, 3.424W | 119 | 109 | 109 |
| | High Muffles | 54.204N, 0.482W | 267 | 130 | 131 |
| | Glen Saugh | 56.542N, 2.333W | 85 | 76 | 80 |
| Hungary | K-Puszta | 46.580N, 19.350E | 125 | 190 | 215 |
| Ireland | Oak Park | 52.527N, 6.552W | 59 | 93 | 93 |
| | Malin Head | 55.223N, 7.203W | 20 | 96 | 96 |
| | Carnsore Point | 52.116N, 6.228W | 9 | 103 | 91 |
| Italy | Montelibretti | 42.600N, 12.380E | 48 | 192 | 165 |
| | Ispra | 45.480N, 8.380E | 29 | 96 | 96 |
| Latvia | Rucava | 56.943N, 21.102E | 18 | 202 | 190 |
| | Zoseni | 57.870N, 25.542E | 188 | 182 | 181 |
| The Netherlands | Kollumerwaard | 53.202N, 6.163E | 1 | 180 | 180 |
| | Vredepeel | 51.322N, 5.511E | 28 | 192 | 204 |
| | De Zilk | 52.180N, 4.300E | 4 | 216 | 216 |





| | Station | Location | Altitude (m) | Number of months available over the observation period (2003-2012) | |
|---|---|---|---|---|---|
| | | | | **Nitrate** | **Ammonium** |
| **Norway** | Birkenes | 58.230N, 8.150E | 190 | 188 | 193 |
| | Skreadolen | 58.490N, 6.430E | 475 | 123 | 123 |
| **Poland** | Jarczew | 51.490N, 21.590E | 180 | 215 | 214 |
| | Sniezka | 50.440N, 15.440E | 1603 | 216 | 216 |
| | Leba | 54.450N, 17.320E | 2 | 216 | 216 |
| **Russia** | Danki | 54.540N, 37.480E | 150 | 149 | 149 |
| **Slovakia** | Chopok | 48.560N, 19.350E | 2008 | 214 | X |
| | Stara Lesna | 49.900N, 20.170E | 808 | 134 | X |
| | Liesek | 49.220N, 19.410E | 892 | 141 | X |
| | Starina | 49.300N, 22.160E | 345 | 226 | 78 |
| | Tropolniky | 47.573N, 17.513E | 113 | 72 | X |
| **Spain** | San Pablo | 39.325N, 4.205W | 917 | X | 78 |
| | Roquetas | 40.491N, 0.292E | 44 | X | 78 |
| | Logroño | 42.272N, 2.301W | 445 | X | 84 |
| | Noia | 42.434N, 8.552W | 683 | X | 75 |
| **Switzerland** | Jungfraujoch | 46.325N, 7.596E | 3578 | 151 | 153 |
| | Payerne | 46.484N, 6.564E | 489 | 61 | 61 |
| | Rigi | 47.430N, 8.275E | 1031 | 61 | X |
| **Turkey** | Cubuk II | 40.300N, 33.000E | 1169 | 121 | 121 |

**Table 2.** Station name, location, altitude and number of months available over the observation period (1994-2014) of the 40 EMEP stations.





| Simulation | REF | NIT | NIT_2 | NIT_3 |
|---|---|---|---|---|
| Ammonium and nitrate aerosols | No | Yes | Yes | Yes |
| Annual cycle of $NH_3$ | / | MACCity | CMIP6 | MACCity |
| Annual cycle of $HNO_3$ | / | Yes | Yes | No (flat) |
| Period | 1979-2016 | | | |

**Table 3.** Configuration of the different simulations in this study. Annual cycles of $NH_3$ and $HNO_3$ of the different simulations are presented in Figure 2 and 3.



| A&N $AOD_{550}$ | Cabauw | Barcelona | Sevastopol |
|---|---|---|---|
| DJF | $0.05 \pm 0.01$ | $0.07 \pm 0.02$ | $0.01 \pm 0.04 \ 10^{-1}$ |
| MAM | $0.17 \pm 0.03$ | $0.15 \pm 0.03$ | $0.05 \pm 0.01$ |
| JJA | $0.11 \pm 0.02$ | $0.09 \pm 0.03$ | $0.03 \pm 0.01$ |
| SON | $0.07 \pm 0.01$ | $0.09 \pm 0.03$ | $0.02 \pm 0.01$ |

**Table 4.** Mean seasonal A&N $AOD_{550}$ simulated by the model (NIT simulation) at three AERONET stations (Cabauw, Barcelona and Sevastopol) over the period 2003-2012.



| AREA | Period | REF | NIT | NIT_2 | NIT_3 | MISR | MODIS Aqua | MODIS Terra |
|---|---|---|---|---|---|---|---|---|
| Europe | DJF | 0.08 ± 0.01 | 0.11 ± 0.01 | 0.11 ± 0.01 | 0.12 ± 0.02 | 0.08 ± 0.01 | 0.08 ± 0.01 | 0.11 ± 0.01 |
| | MAM | 0.10 ± 0.02 | 0.21 ± 0.01 | 0.20 ± 0.01 | 0.19 ± 0.01 | 0.15 ± 0.03 | 0.18 ± 0.04 | 0.21 ± 0.04 |
| | JJA | 0.10 ± 0.03 | 0.18 ± 0.02 | 0.19 ± 0.03 | 0.18 ± 0.02 | 0.16 ± 0.03 | 0.21 ± 0.02 | 0.24 ± 0.04 |
| | SON | 0.09 ± 0.02 | 0.14 ± 0.02 | 0.15 ± 0.02 | 0.15 ± 0.02 | 0.11 ± 0.03 | 0.12 ± 0.03 | 0.15 ± 0.04 |
| | **Overall** | **0.09 ± 0.02** | **0.16 ± 0.01** | **0.16 ± 0.01** | **0.16 ± 0.01** | **0.13 ± 0.02** | **0.16 ± 0.02** | **0.19 ± 0.03** |
| Mediterranean Sea | DJF | 0.16 ± 0.05 | 0.17 ± 0.05 | 0.17 ± 0.05 | 0.17 ± 0.05 | 0.15 ± 0.03 | 0.15 ± 0.02 | 0.15 ± 0.02 |
| | MAM | 0.29 ± 0.08 | 0.34 ± 0.08 | 0.34 ± 0.08 | 0.34 ± 0.08 | 0.22 ± 0.03 | 0.25 ± 0.03 | 0.27 ± 0.03 |
| | JJA | 0.16 ± 0.03 | 0.19 ± 0.03 | 0.20 ± 0.03 | 0.19 ± 0.03 | 0.24 ± 0.05 | 0.23 ± 0.03 | 0.25 ± 0.04 |
| | SON | 0.15 ± 0.03 | 0.17 ± 0.03 | 0.18 ± 0.03 | 0.17 ± 0.03 | 0.19 ± 0.03 | 0.18 ± 0.03 | 0.20 ± 0.03 |
| | **Overall** | **0.19 ± 0.03** | **0.22 ± 0.02** | **0.22 ± 0.02** | **0.22 ± 0.02** | **0.20 ± 0.03** | **0.20 ± 0.02** | **0.22 ± 0.02** |

**Table 5.** Seasonal averages and annual mean total $AOD_{550}$ simulated by ALADIN-Climate for NIT and REF configurations and measured by MODIS and MISR over the period 2001-2016 (2003-2016 for MODIS Aqua).





| AOD$_{550}$ | MISR | MODIS Aqua | MODIS Terra |
|---|---|---|---|
| NIT | 0.90 (0.04) | 0.75 (0.02) | 0.77 (- 0.02) |
| NIT_2 | 0.85 (0.04) | 0.88 (0.01) | 0.89 (- 0.02) |
| NIT_3 | 0.95 (0.03) | 0.82 (0.01) | 0.85 (- 0.02) |

**Table 6.** Temporal correlation (and bias), over Europe, between mean annual cycles as plotted in Figure 6 from simulations (NIT, NIT_2 and NIT_3) and satellite products (MISR, MODIS Aqua and MODIS Terra) over the period 2001-2016 (2003-2016 for MODIS Aqua).





| AOD$_{550}$ **trend per decade (1979-2016)** | NIT | | | NAB2013 |
|---|---|---|---|---|
| | Total | A&N | Sulphate | |
| Europe | 0.10 | 0.21 | -0.046 | -0.045 |
| Mediterranean Sea | 0.10 | 0.21 | -0.046 | -0.045 |

**Table 7.** NIT simulation AOD$_{550}$ trend per decade compared to NAB2013 (Nabat et al., 2013) over the period 1979-2016.



| AOD$_{550}$ trend per decade (2003-2015) | NIT | | | | NAB2013 | MISR | MODIS A | MODIS T |
|---|---|---|---|---|---|---|---|---|
| | Total | A&N | Dust | Sulphate | | | | |
| Europe | -0.006 | 0.013 | 0.002 | -0.022 | -0.015 | -0.022 | -0.029 | -0.019 |
| Mediterranean Sea | -0.010 | 0.011 | -0.002 | -0.018 | -0.011 | -0.035 | -0.029 | -0.028 |

**Table 8.** NIT simulation AOD$_{550}$ trend per decade compared to NAB2013 (Nabat et al., 2013) and total AOD trend from different satellite products (MISR, MODIS Aqua and MODIS Terra) over the period 2003-2015.





| SW DRF (W m$^{-2}$) | REF | NIT | CNRM-RCSM4 | MODIS |
|---|---|---|---|---|
| | This work | | Nabat et al. (2015b) | Papadimas et al. (2012) |
| Surface | -7.6 | -8.7 | -19.9 | -16.5 |
| TOA | -1.5 | -2.4 | -7.8 | -2.4 |

**Table 9.** Aerosol SW DRF averages (W m$^{-2}$) in all sky conditions at the surface and at TOA for different datasets over the Mediterranean basin (29-46.5°N, 10.5°W-38.5°E).



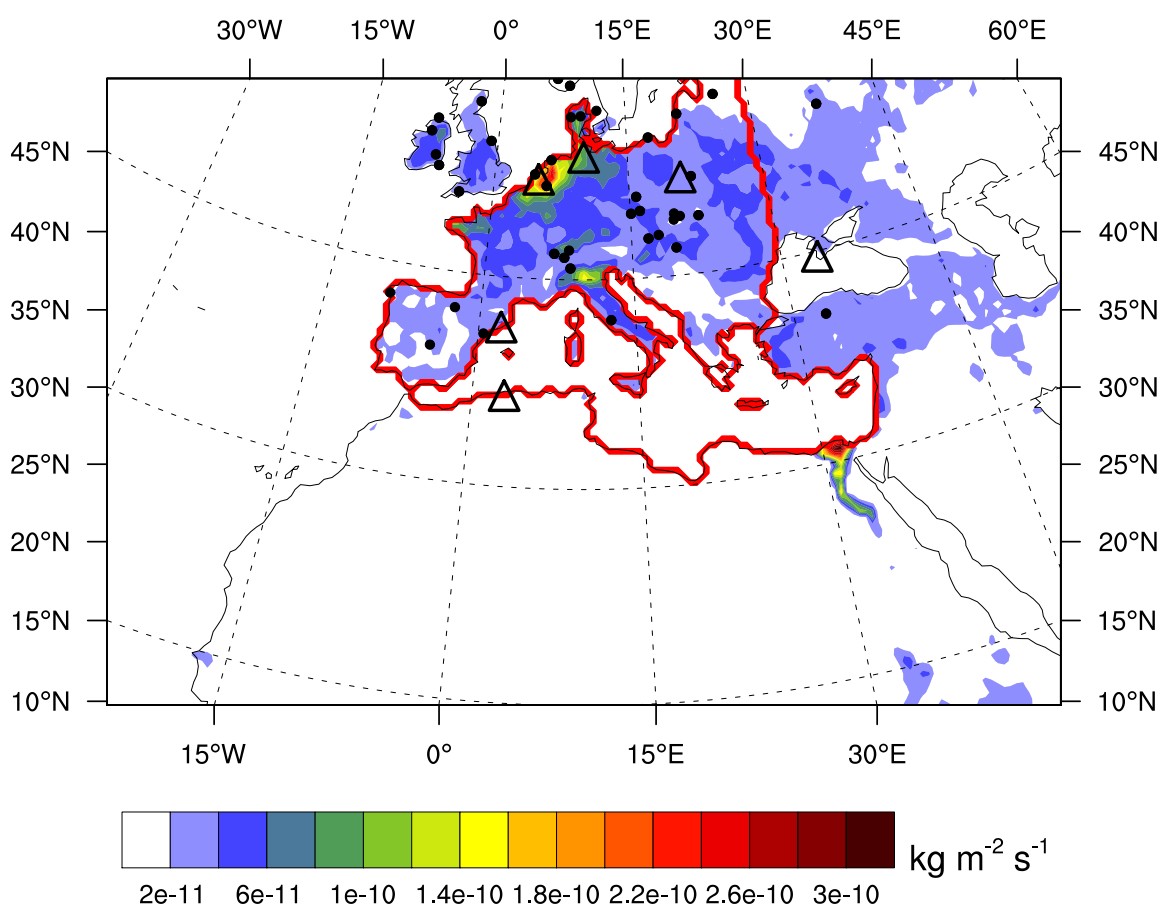

**Figure 1.** NH$_3$ emissions (CMIP6) over the period 1979-2016. The inner model domain represents 101x153 points (without the bi-periodization and the relaxation zone). The different zones studied (Europe, Mediterranean) are represented in red. The observation data is symbolized by a black dot (EMEP stations) and by a black triangle (AERONET stations). The projection type used here is the Lambert conformal projection.





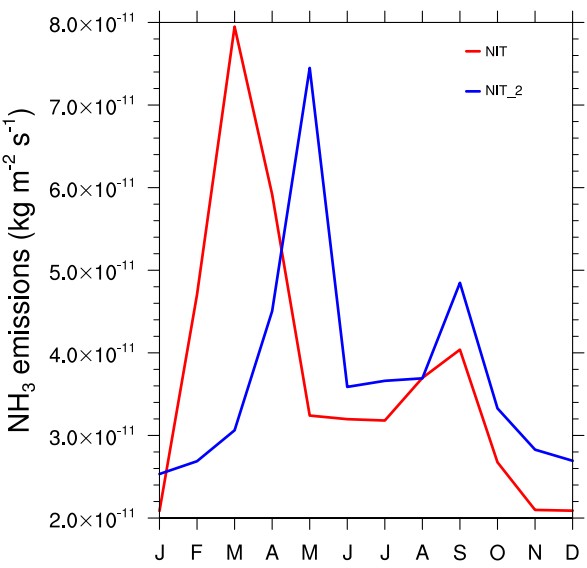

**Figure 2.** 1979-2016 monthly means of $NH_3$ emissions (kg m$^{-2}$ s$^{-1}$) over Europe as defined in Figure 1, used for NIT (red, based on MACCity) and NIT_2 (blue, from CMIP6) simulations. The annual total is equal in the two versions (4.5 10$^{-10}$ kg m$^{-2}$ s$^{-1}$).



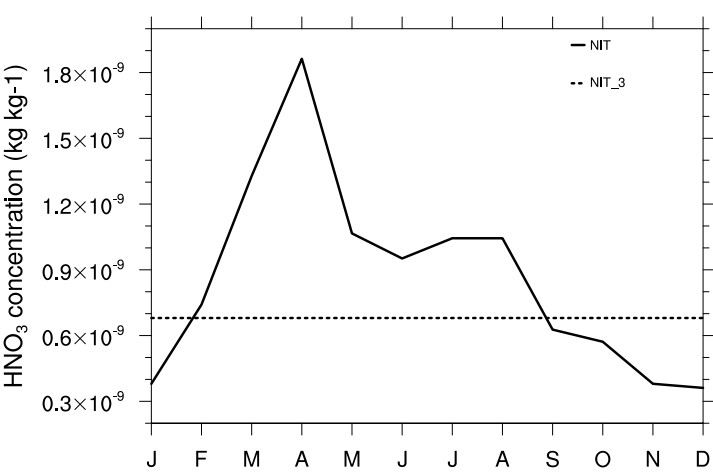

**Figure 3.** 2003-2007 monthly mean of HNO₃ surface concentration over Europe (as defined in Figure 1).





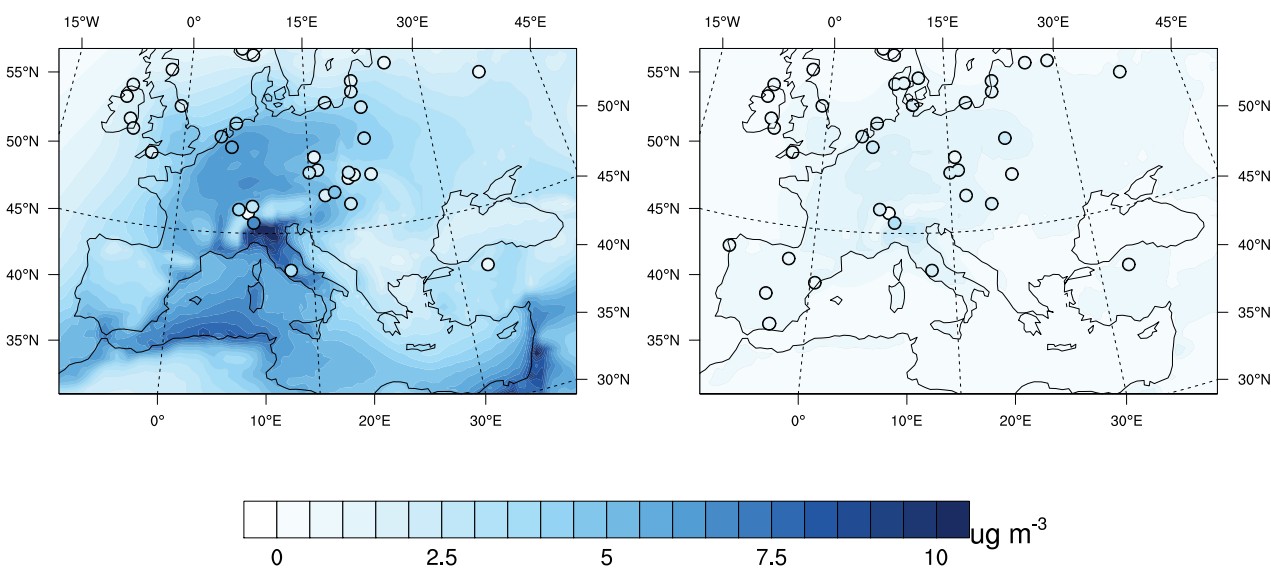

**Figure 4.** Nitrate (left) and ammonium (right) surface concentration (µg m$^{-3}$) simulated by the ALADIN-Climate model (1994-2014) and measured at EMEP stations (coloured dots) using the same color palette.





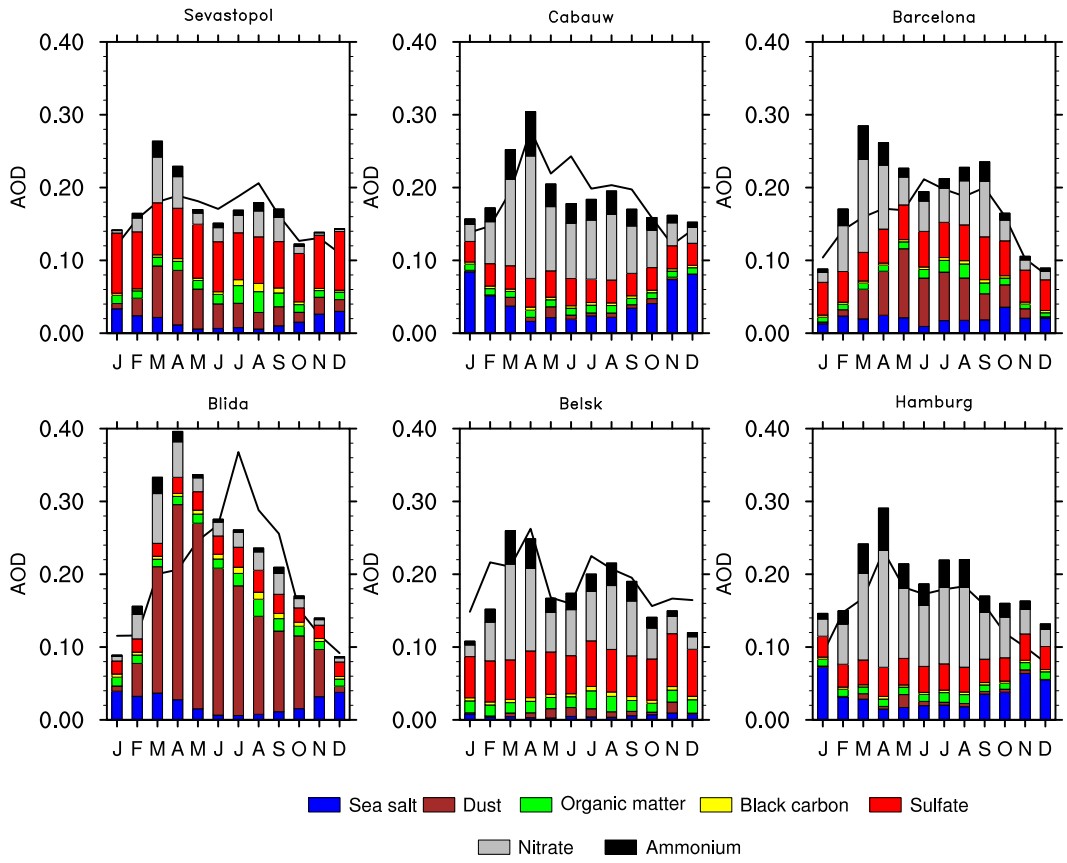

**Figure 5.** Comparison of the average annual cycle (2003-2012) of aerosol optical depth (at 550 nm) simulated by the ALADIN-Climate model (bars), with measurements at six selected AERONET stations (line). The model contribution of each aerosol type is indicated using different color bars.







**Figure 6.** Total aerosol optical depth (at 550 nm) simulated by the ALADIN-Climate model for the NIT and REF simulations and measured from MODIS and MISR, averaged over the period 2001-2016 (2003-2016 for MODIS Aqua).

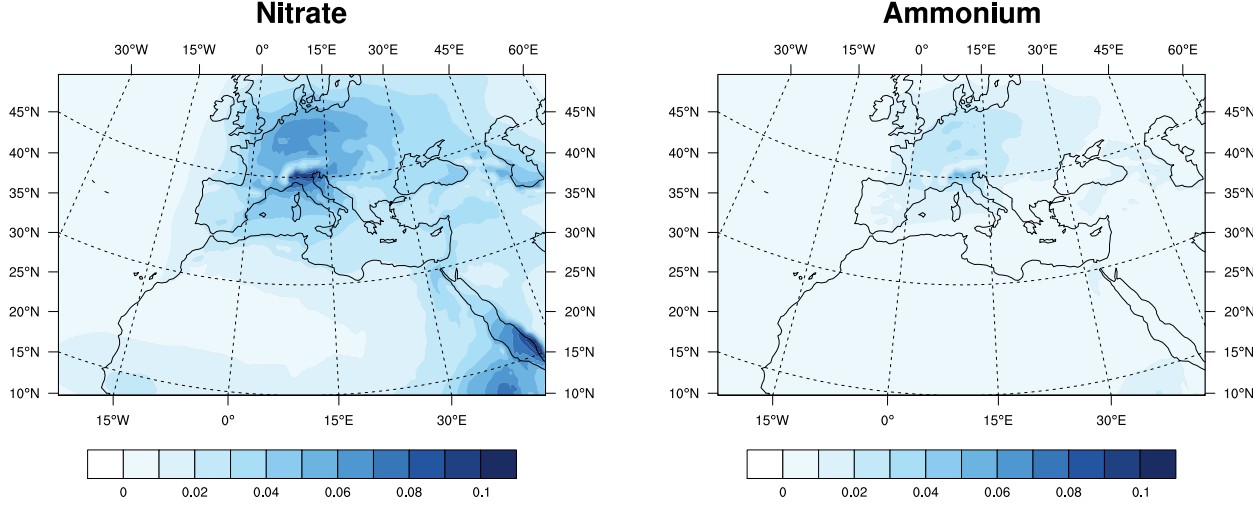

**Figure 7.** Optical depth of nitrate (left) and ammonium (right) at 550 nm simulated by ALADIN-Climate over the period 2001-2016 (NIT simulation).



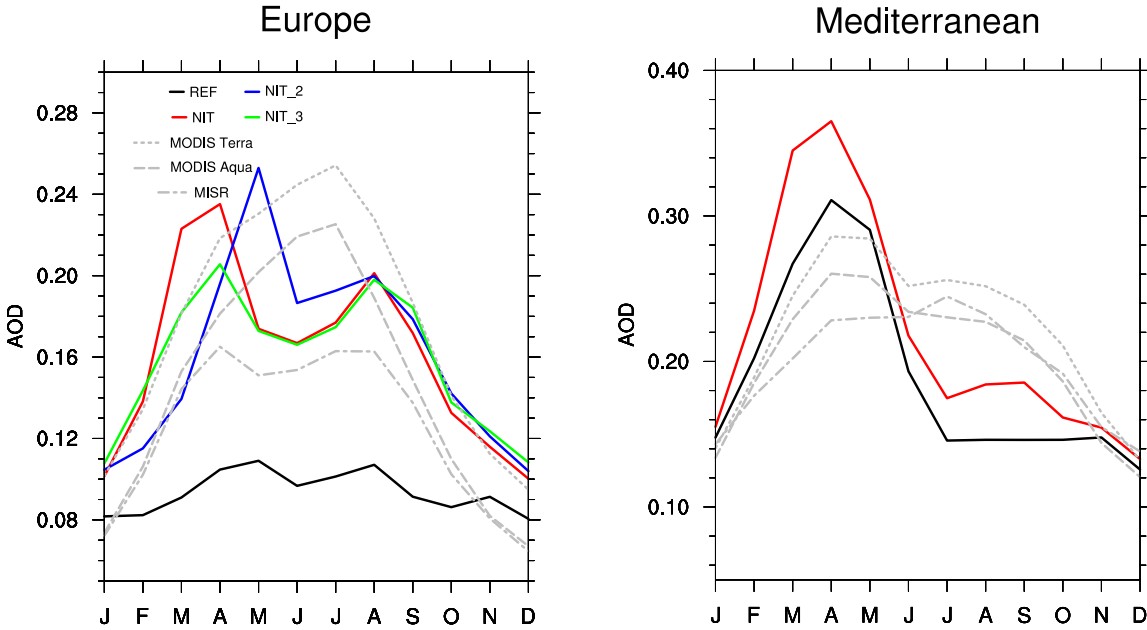

**Figure 8.** Average total aerosol AOD$_{550}$ annual cycle simulated by ALADIN-Climate, over the period 2001-2016 (2003-2016 for MODIS Aqua), with (NIT) and without (REF) A&N and measured by two satellite instruments (MODIS and MISR) over Europe (left) and Mediterranean Sea (right).





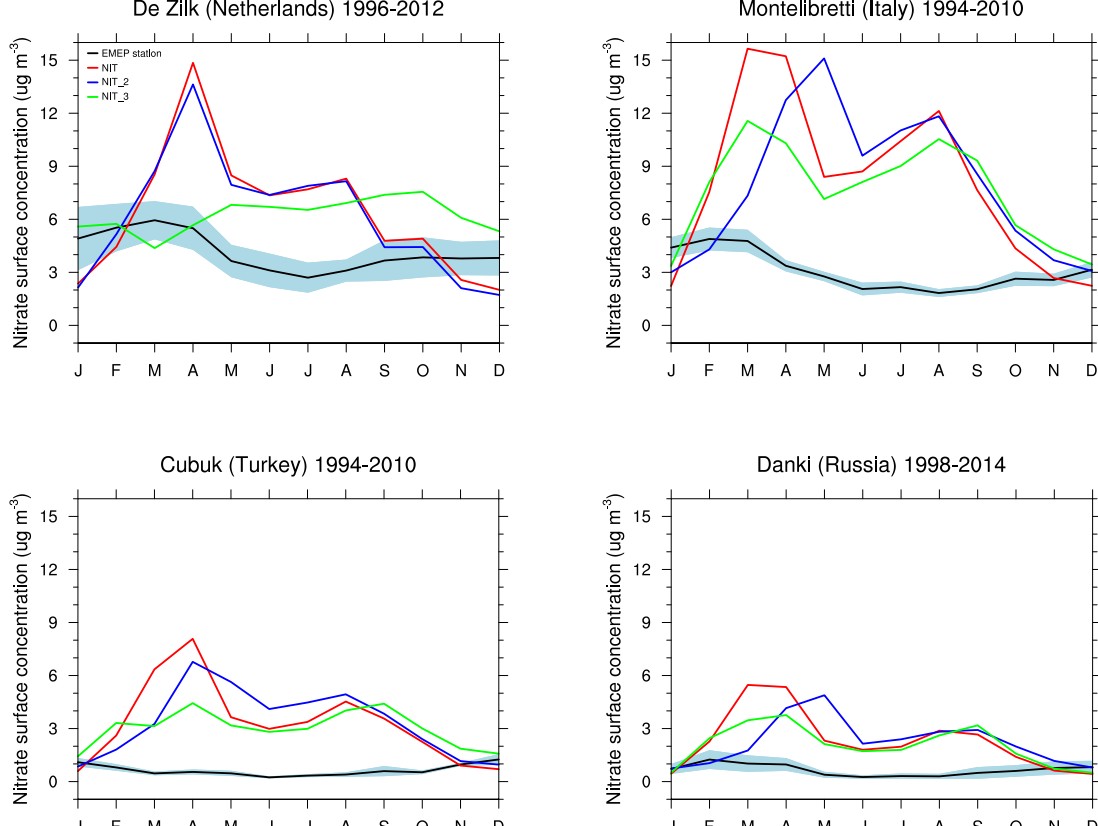

**Figure 9.** Monthly means of the surface nitrate concentration (µg m$^{-3}$) at the De Zilk, Montelibretti, Cubuk and Danki EMEP stations, for NIT (red), NIT_2 (blue) and NIT_3 (green) simulations. Observations are in black and the standard deviation associated with observations is in light blue.





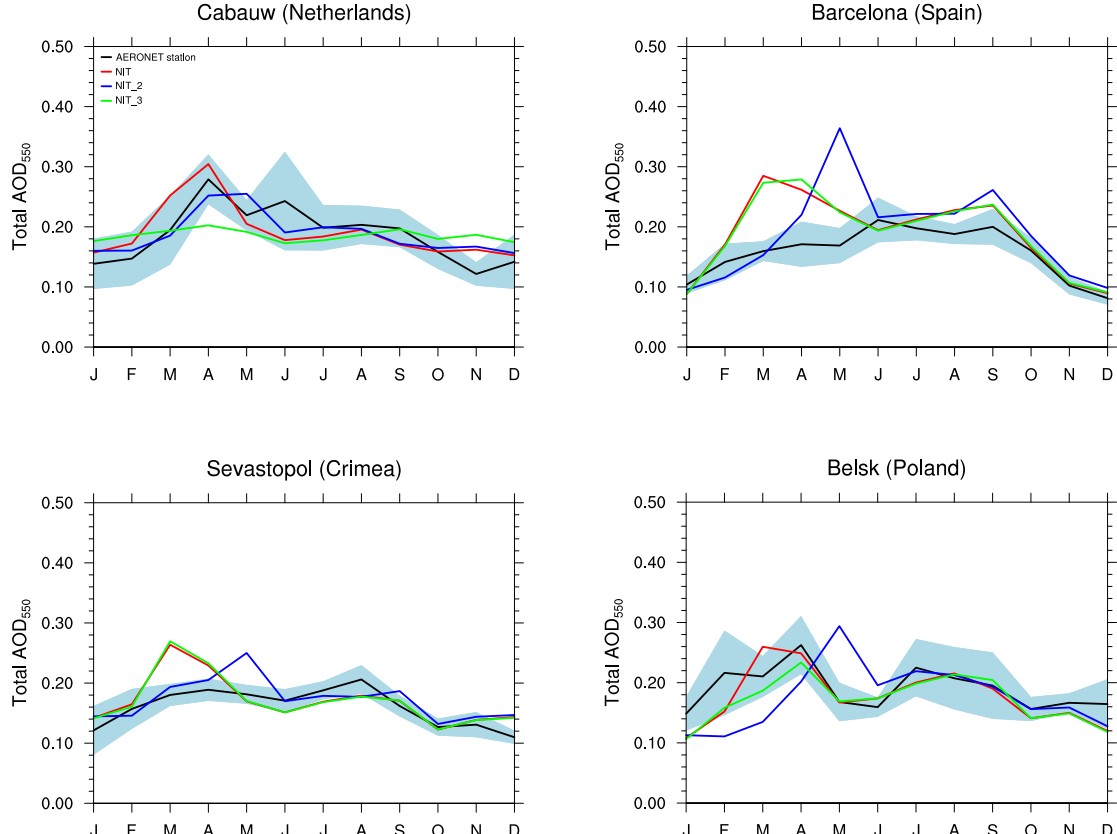

**Figure 10.** Comparison of the monthly means (2003-2012) of aerosol optical depth (at 550 nm) simulated by the ALADIN-Climate model, with measurements at four AERONET stations for NIT (red), NIT_2 (blue) and NIT_3 (green) simulations. Observations are in black and the standard deviation associated with observations is in light blue.





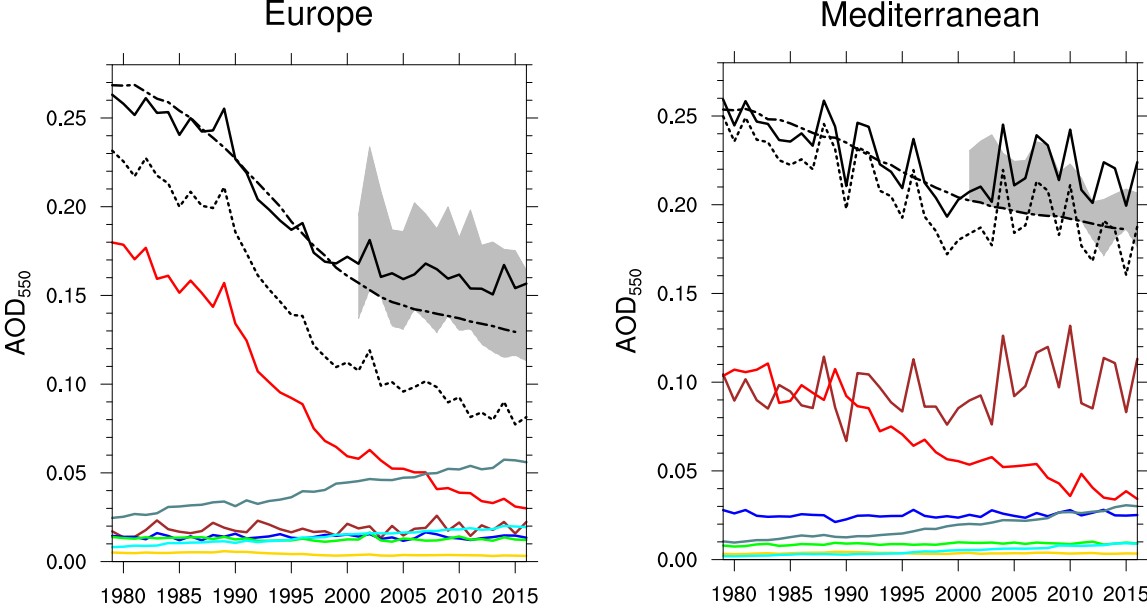

**Figure 11.** Yearly means of $AOD_{550}$ over Europe (left) and the Mediterranean sea (right) as defined in Figure 1 for the NIT simulation (solid black line), the REF simulation (dotted black line), the NAB2013 climatology (dotted dashed black line) and for each aerosol species in NIT (solid colour lines) over the period 1979-2016 compared to satellite products (grey area from 2001 on). Dust in brown, sea-salt in blue, black carbon in gold, organic carbon in green, sulphate in red, ammonium in cyan and nitrate in sea-green.




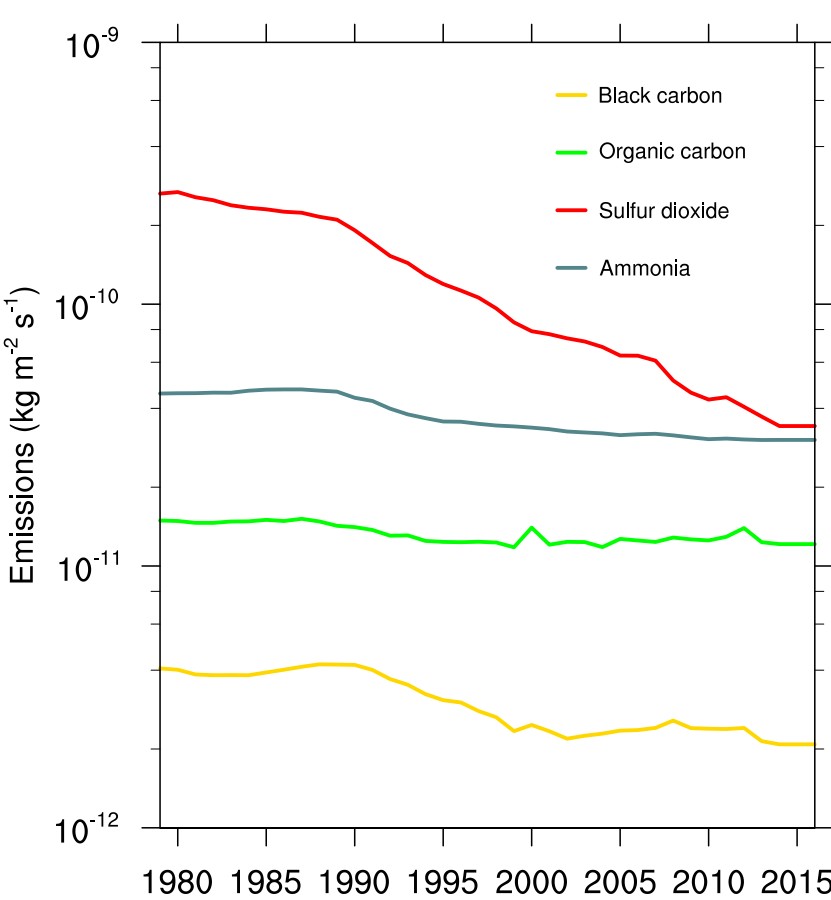

**Figure 12.** Yearly means of anthropogenic emissions over Europe as defined in Figure 1 for black carbon (yellow), organic carbon (green), sulphur dioxide (red) and ammonia (blue-grey) over the period 1979-2016.



**Figure 13.** Clear-sky (top) and all-sky (bottom) SW direct radiative forcing (W m$^{-2}$) due to A&N aerosols at the surface (left) and at the top of the atmosphere, TOA (right) estimated over the period 1979-2016 by difference between the REF and NIT simulations.



**Figure 14.** Yearly means of aerosol DRF over Europe (left) and the Mediterranean sea (right) for nitrate and ammonium (cyan), sulphate and organics (red), black carbon (yellow), dust (brown), sea-salt (blue) and total (black) in all sky (solid lines) and clear sky (dotted lines) conditions over the period 1979-2016.





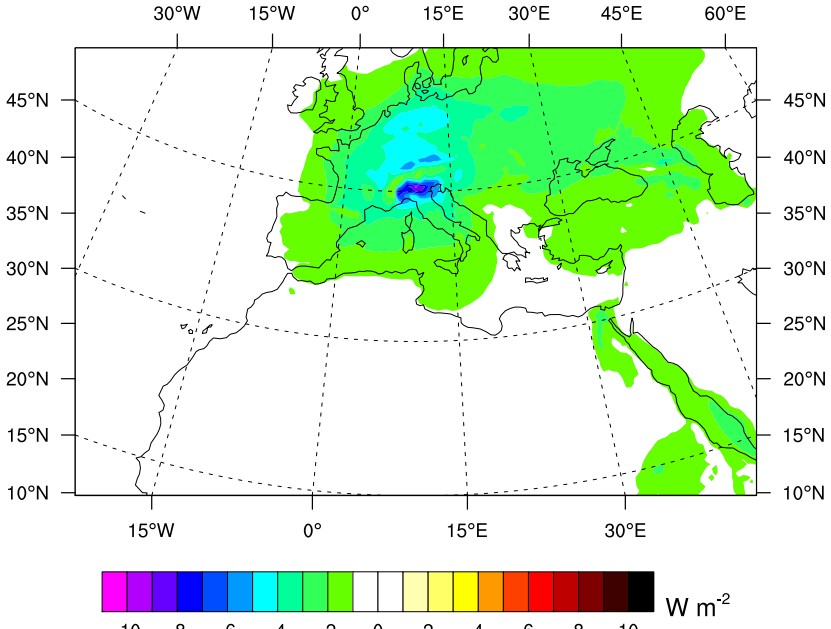

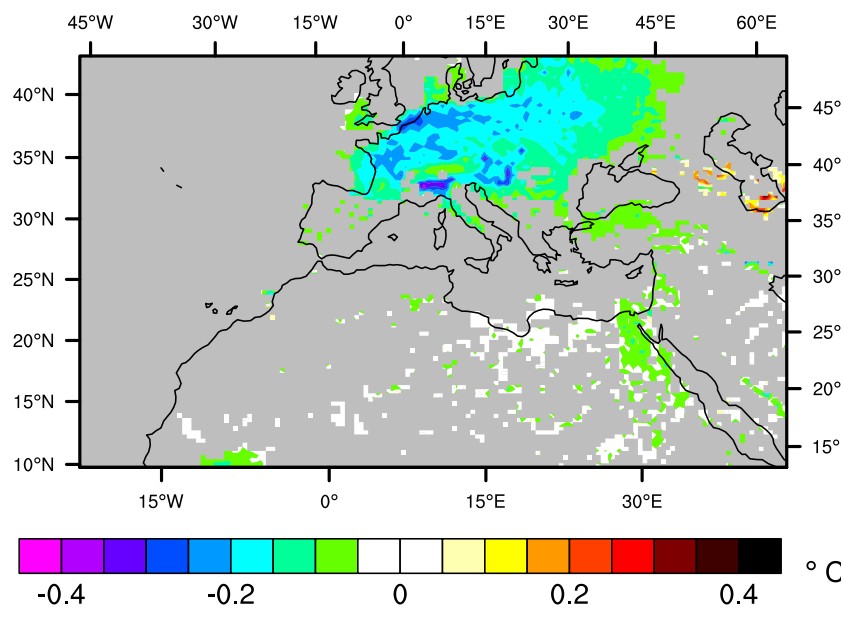

**Figure 15.** DRF SW (W m$^{-2}$, top) at the surface in all sky conditions and the A&N aerosols impact on the near-surface air temperature at 2 m (°C, bottom) on average over the period 1979-2016 in summer (JJA) obtained by difference between the REF and NIT simulations. For the temperature map, the grey area is not statistically significant at the 0.05 level.