# Peer review of "Model simulation of ammonium and nitrate aerosols distribution in the Euro-Mediterranean region and their radiative and climatic effects over 1979-2016"

_Atmospheric Chemistry and Physics, 2018_

## Editor Comment (EC1) · Dulac (Editor) · 13 Dec 2018

Thanks for correcting your initial version. I suggest that you consider the following points for further revision :

- Could you apportion ammonium between nitrate and sulphate? Did you check if the introduction of ammonium impact the sulphate content in the NIT simulation compared to the REF?

- I find that you somehow miss to relate the behaviour of the model at the surface and

in the column. Even if times series were more limited, it could be informative to look for as close as possible places with simultaneous AERONET and EMEP data in order to check concurrently the model at the surface and in the column. If there were no such interesting possibility from the datasets, it would be worth stating it.

- The title of Section 4 ("Evaluation of the new AN aerosol scheme") indicates that it is dedicated to evaluating the new model aerosol scheme; but it also includes analyses of the aerosol distribution and trends that I believe go well beyond the simple model evaluation; according to me, to better valorise your results regarding the distribution of these aerosols, it is worth reorganizing this section in order to make a new one following the evaluation section, that will be dedicated to the analysis of the spatio-temporal variability and trends of ammonium and nitrate aerosols, and their impact on particulate air quality (not only their direct radiative impact and related effects on climatic parameters); for instance, Figure 2 could be completed with maps of surface PM (e.g. PM1 at least) for discussing the relative contributions from the A&N species.

---

## Referee Comment (RC1) · Anonymous Referee #1 · 2 Jan 2019

This is an interesting regional modelling study which investigates the simulated ammonium and nitrate aerosols distribution over Europe and Mediterranean region and their radiative and climatic effects over the period 1979-2016. In general, it is well written and structured and there are original model results presented and discussed. Please see below a few comments that have to be taken into consideration before acceptance of the manuscript for publication.

Comments 1) Page 3, lines 3-4: Recent studies indicate that dust may play a very important role in nitrate formation, e.g. Karydis et al. (ACP, 2017) suggest that the

tropospheric burden of aerosol nitrate increases by 44% when interactions of nitrate with mineral dust are considered. 2) The authors write that in the present simulations, aerosols are not included in the lateral boundary forcing because the domain is supposed to be large enough to include all the sources of aerosols affecting the Mediterranean region. In this case Mediterranean region could be affected by dust aerosols advected from Sahara and Middle East. Please comment if the dust source regions are within the selected domain. A similar issue for the selection of the domain and extension south of the Sahara is discussed by Tsikerdekis et al. (ACP, 2017) in order to reduce the contribution of dust aerosols from outside the domain. 3) The authors use 3 dust bins (0.01 to 1.0, 1.0 to 2.5 and 2.5 to 20 $\mu$m) for their experiments. Foret et al. (JGR, 2006) suggests that dust size for the transported bins should range from 0.09 to 63 $\mu$m, considering both the total number and the mass distribution of soil particles. Please briefly discuss how sensitive could be the model results with the number and range of size bins? 4) Please provide some references for the selection and use of the proposed values on the deposition velocities? 5) Page 8, lines 30-31: The authors may also consider a recent article that focuses on the differences between the MODIS Collection 6 and 5.1 aerosol datasets over the greater Mediterranean region (Georgoulias et al., Atmos. Env., 2016). Details about the different MODIS algorithms and the corresponding uncertainties can be found there. The authors may consider to replace $\pm 0.03$ with the values given there for each algorithm separately and also mention that they use the combined DT+DB aerosol product. 6) Page 15, lines 33-34: The authors mention that "An interesting point is that the nitrate AOD550 rise is not due to an increase in its precursors (ammonia and nitric acid)." It might be interesting to compare maps of nitrate trends per decade with recent satellite-based trends on tropospheric NO2 which show negative trends during the last 10-20 years (e.g. Hilboll et al., ACP, 2013; Georgoulias et al., ACPD, 2018) over the western part of the Euro-Mediterranean domain and positive over the eastern part. 7) The authors refer to article of Zanis (2009). However, there is also a more robust study with a 12-years simulation (Zanis et al., Climate Research, 2012) indicating a limited direct shortwave effect of anthropogenic aerosols

(carbonaceous and sulphate) on the regional European climate with the greatest negative temperature difference of $-0.2°C$ over the Balkan Peninsula. 8) An issue that it is not discussed at all is if the aerosol induced signal on the temperature fields is higher than the model's internal variability. I think at least a few comments on this issue are necessary. This is also a part of the limitations in such simulations. 9) The author write in line 11 of page 3 " Nevertheless, the predicted trend in surface nitrate . . ." Does this refer to particulate nitrate?

Technical comments 1) Page 2, lines 21-23: Please consider rephrasing. e.g. "After the small particles reach equilibrium . . ." 2) Page 5, line 24: Please rephrase, accordingly. I guess the authors mean that there is no change from year to year in the annual cycle of HNO3. 3) The equations should be numbered within the text. 4) Page 7, line 9: [NH4] = TA - [NH3]. In accordance with the previous notation NH4 should be NH4 cations. 5) Page 8, line 12-13: Please add the units for the values 4.3, 34.9, 4.8 and 34.2. 6) Page 10, line 31: I think it should be Figure 4 instead of Figure 2. 7) Page 12, line 4: It should rather read ". . . some biases are identified . . ." instead of ". . . some bias are identified . . ." 8) Page 12, line 18: I think it should be Figure 6 instead of Figure 4.

---

## Referee Comment (RC2) · Anonymous Referee #2 · 22 Jan 2019

The authors present work on the implementation and the evaluation of a simplified ammonium and nitrate module in the TACTIC aerosol scheme, to be utilized in the ALADIN regional climate model and to assess the direct radiative effect and climatic impact of the ammonium and nitrate aerosols over the Euro-Mediterranean region. In general, the work presented covers a scientifically sound topic, is well written, structured and the results included are original and discussed. Please find below several comments that can be taken into account before the manuscript is accepted for publication.

1) I cannot find any explanation on the statistical figures used for the model validation

against the different observational sources included in the study. Further discussion should be elaborated on the validation methodology.

2) The limited number of bins can hamper the interpretation of results. Especially for dust, the coarsest bin ranges from 2.5 to 20 micrometers. Because of the domain of study, including several important sources of dust emissions, a large fraction of particles are excluded from the analysis. Further discussion should be devoted to the selection of the bins and why a range like that has been selected.

3) Page 9, Line 25 "These 25 stations do not have continuous data over the period 1994-2014 and we selected those with a minimum of 5 years of data, which may be non-continuous but a minimum of 5 observations is necessary for every month.". I think that 5 observations is not representative of a monthly average. How is the comparison with model results done? Do you estimate a monthly average from those 5 observations? Or do you extract model data for those 5 time steps and then compare with observations?

4) Are the evaluation results different for MODIS and AERONET at AERONET sites? Or, better side, is there any significant difference when evaluating the model against satellite and AERONET in those locations where you have AERONET observations?

5) Can you do some partitioning between ammonia and nitrate? Can you somehow estimate which is the individual contribution of each component to the direct radiative forcing?

Last, it would help readability if equations are numbered within the manuscript.

---

## Author Comment (AC1) · 1 Mar 2019

Thanks for correcting your initial version. I suggest that you consider the following points for further revision:
I would like to thank François Dulac for his interactive comment which mentions different points listed below.

Could you apportion ammonium between nitrate and sulphate? Did you check if the

[Figure]

introduction of ammonium impact the sulphate content in the NIT simulation compared to the REF?

In the ALADIN-Climate model, ammonium sulphate aerosols are not affected by ammonium nitrate production because ammonium nitrate aerosols are formed only after sulphates in case NH3 has not been entirely consumed by sulfate formation. The formation of ammonium sulphate takes priority over ammonium nitrate formation due to the low vapor pressure of sulfuric acid (Hauglustaine et al. 2014).

I find that you somehow miss to relate the behaviour of the model at the surface and in the column. Even if times series were more limited, it could be informative to look for as close as possible places with simultaneous AERONET and EMEP data in order to check concurrently the model at the surface and in the column. If there were no such interesting possibility from the datasets, it would be worth stating it.

The area around Cabauw is represented by one AERONET station (Cabauw) and two EMEP stations (De Zilk and Vredepeel; 60 and 100 km respectively). At this site, there are a total aerosol AOD and a surface A&N concentration that are well reproduced by the model. A sentence was added in the article to show the good consistency between those two parameters over this area (section 4.2).

The title of Section 4 ("Evaluation of the new AN aerosol scheme") indicates that it is dedicated to evaluating the new model aerosol scheme; but it also includes analyses of the aerosol distribution and trends that I believe go well beyond the simple model evaluation; according to me, to better valorise your results regarding the distribution of these aerosols, it is worth reorganizing this section in order to make a new one following the evaluation section, that will be dedicated to the analysis of the spatio-temporal variability and trends of ammonium and nitrate aerosols, and their impact on particulate air quality (not only their direct radiative impact and related effects on climatic parameters); for instance, Figure 2 could be completed with maps of surface

PM (e.g. PM1 at least) for discussing the relative contributions from the A&N species. To better valorise the results, the title of Section 4 has been changed by "Evaluation of the new aerosol scheme and analysis of the spatio-temporal variability and trends of A&N aerosols". Concerning the impact of aerosols on particulate air quality, our model is not very suitable for answering this question. Indeed, PM1 is not a diagnostic and anthropogenic aerosols do not have speciation in size but a bulk approach with 1 bin for sulphate particles and 2 bins (hydrophilic and hydrophobic particles) for organics and for black carbon.

---

## Author Comment (AC2) · 1 Mar 2019

This is an interesting regional modelling study which investigates the simulated ammonium and nitrate aerosols distribution over Europe and Mediterranean region and their radiative and climatic effects over the period 1979-2016. In general, it is well written and structured and there are original model results presented and discussed. Please see below a few comments that have to be taken into consideration before acceptance of the manuscript for publication.

I would like to thank the anonymous referee for his comments which mention different

points listed below.

Page 3, lines 3-4: Recent studies indicate that dust may play a very important role in nitrate formation, e.g. Karydis et al. (ACP, 2017) suggest that the tropospheric burden of aerosol nitrate increases by 44% when interactions of nitrate with mineral dust are considered.
A sentence has been added to complete the introduction and mention this article (section 1).

The authors write that in the present simulations, aerosols are not included in the lateral boundary forcing because the domain is supposed to be large enough to include all the sources of aerosols affecting the Mediterranean region. In this case Mediterranean region could be affected by dust aerosols advected from Sahara and Middle East. Please comment if the dust source regions are within the selected domain. A similar issue for the selection of the domain and extension south of the Sahara is discussed by Tsikerdekis et al. (ACP, 2017) in order to reduce the contribution of dust aerosols from outside the domain.
The area is large enough to integrate the two main sources of dust, which are the Sahara (more precisely the Bodélé and the area covering eastern Mauritania, western Mali and southern Algeria) and the largest part of the Arabian Peninsula. The dust sources further south are not taken into account because they have no impact over the studied area (e.g. Moulin et al., 1998).

The authors use 3 dust bins (0.01 to 1.0, 1.0 to 2.5 and 2.5 to 20) for their experiments. Foret et al. (JGR, 2006) suggests that dust size for the transported bins should range from 0.09 to 63, considering both the total number and the mass distribution of soil particles. Please briefly discuss how sensitive could be the model results with the number and range of size bins?

We only use 3 dust bins to keep a model with a relatively low computational cost. We are aware that this is a limit for our study. Indeed, Foret et al. (JGR, 2006) indicate that dust size for the transported bins should range from 0.09 to 63 and at least eight size bins are necessary to secure an 8% accuracy on the total suspended dust mass after two days of transport. This limit must therefore be taken into account. On the other hand, our model approaches the LMDz-INCA (from which our nitrate scheme originates) in term of number of bin because they also use 3 ranges (<1 , 1-10 and >10. A sentence has been added in the article to inform of this limit (section 2.2.3).

Please provide some references for the selection and use of the proposed values on the deposition velocities?
For fine nitrates, as Hauglustaine et al., 2014, we used deposition values close to sulphates, already used in the model. For coarse nitrates, the values close to coarse dust and sea-salt aerosols have been used. All data are detailed in Michou et al. 2015 (Table 1). The reference has been added to the article (section 2.2.4).

Page 8, lines 30-31: The authors may also consider a recent article that focuses on the differences between the MODIS Collection 6 and 5.1 aerosol datasets over the greater Mediterranean region (Georgoulias et al., Atmos. Env., 2016). Details about the different MODIS algorithms and the corresponding uncertainties can be found there. The authors may consider to replace $\pm$ 0.03 with the values given there for each algorithm separately and also mention that they use the combined DT+DB aerosol product.
These details and this reference have been added to the article (section 3.1).

Page 15, lines 33-34: The authors mention that "An interesting point is that the nitrate AOD550 rise is not due to an increase in its precursors (ammonia and nitric acid)." It might be interesting to compare maps of nitrate trends per decade with recent

satellite-based trends on tropospheric NO2 which show negative trends during the last 10-20 years (e.g. Hilboll et al., ACP, 2013; Georgoulias et al., ACPD, 2018) over the western part of the Euro-Mediterranean domain and positive over the eastern part.

It is indeed interesting to notice these trends in NO2, but a detailed study of the trends of this species is out of the scope of this paper. Howerver a sentence mentioning the decline in tropospheric NO2 over the western part of the Euro-Mediterranean domain and these two references have been added to the article (section 4.6).

The authors refer to article of Zanis (2009). However, there is also a more robust study with a 12-years simulation (Zanis et al., Climate Research, 2012) indicating a limited direct shortwave effect of anthropogenic aerosols (carbonaceous and sulphate) on the regional European climate with the greatest negative temperature difference of -0.2C over the Balkan Peninsula.

This result and the reference were added to the article (section 5.2).

An issue that it is not discussed at all is if the aerosol induced signal on the temperature fields is higher than the model's internal variability. I think at least a few comments on this issue are necessary. This is also a part of the limitations in such simulations.

A t-test has been done using the 38 years of the simulation with a significant level of 95%. Furthermore, areas with high temperature differences are areas with high A&N AOD so it is unlikely to be due to model's internal variability. These details have been added to the article (section 5.2).

The author write in line 11 of page 3 "Nevertheless, the predicted trend in surface nitrate ..." Does this refer to particulate nitrate?

Here, this refer to nitrate concentration at the surface. A clarification has been brought to the text (section 1).

Technical comments 1) Page 2, lines 21-23: Please consider rephrasing. e.g. "After the small particles reach equilibrium ..." 2) Page 5, line 24: Please rephrase, accordingly. I guess the authors mean that there is no change from year to year in the annual cycle of HNO3. 3) The equations should be numbered within the text. 4) Page 7, line 9: [NH4] = TA - [NH3]. In accordance with the previous notation NH4 should be NH4 cations. 5) Page 8, line 12-13: Please add the units for the values 4.3, 34.9, 4.8 and 34.2. 6) Page 10, line 31: I think it should be Figure 4 instead of Figure 2. 7) Page 12, line 4: It should rather read "... some biases are identified ..." instead of "... some bias are identified ..." 8) Page 12, line 18: I think it should be Figure 6 instead of Figure 4. Done.

---

## Author Comment (AC3) · 1 Mar 2019

The authors present work on the implementation and the evaluation of a simplified ammonium and nitrate module in the TACTIC aerosol scheme, to be utilized in the ALADIN regional climate model and to assess the direct radiative effect and climatic impact of the ammonium and nitrate aerosols over the Euro-Mediterranean region. In general, the work presented covers a scientifically sound topic, is well written, structured and the results included are original and discussed. Please find below several comments that can be taken into account before the manuscript is accepted

for publication.

I thank the second anonymous referee for his reviewer's comment which mentions different points listed below.

I cannot find any explanation on the statistical figures used for the model validation against the different observational sources included in the study. Further discussion should be elaborated on the validation methodology.

In Figure 9 and Figure 10, we have calculated the confidence interval (light blue spread), associated with observations, using the number of years available for each station with a significant level of 95%. A sentence has been added in the text to mention that (section 4.5).

The limited number of bins can hamper the interpretation of results. Especially for dust, the coarsest bin ranges from 2.5 to 20 micrometers. Because of the domain of study, including several important sources of dust emissions, a large fraction of particles are excluded from the analysis. Further discussion should be devoted to the selection of the bins and why a range like that has been selected.

We only use 3 dust bins to keep a simple model with a relatively low computational cost. We are aware that excluding the coarsest dust particles is a limit for our study and a sentence has been added in the article to inform of this limit (section 2.2.3). Moreover the dust affecting Europe and the Mediterranean are mainly fine aerosols (coarse particles will fall quickly) and are therefore taken into account.

Page 9, Line 25 "These 25 stations do not have continuous data over the period 1994-2014 and we selected those with a minimum of 5 years of data, which may be non-continuous but a minimum of 5 observations is necessary for every month.". I think that 5 observations is not representative of a monthly average. How is the comparison with model results done? Do you estimate a monthly average from those

5 observations? Or do you extract model data for those 5 time steps and then compare with observations?

The sentence may not be clear enough but it is monthly data. We select only the stations with at least 5 data for each month (so with at least 5 months of January, 5 months of February ...). A new sentence has been added to the article (section 3.1).

Are the evaluation results different for MODIS and AERONET at AERONET sites? Or, better side, is there any significant difference when evaluating the model against satellite and AERONET in those locations where you have AERONET observations?

Figure 5 has been redone by adding the AOD annual cycle of the different satellites (MISR, MODIS Aqua and MODIS Terra). A comparison between AERONET and the different satellites has been added to the text (section 4.2).

Can you do some partitioning between ammonia and nitrate? Can you somehow estimate which is the individual contribution of each component to the direct radiative forcing?

Ammonium represents about 35% of the nitrate AOD over Europe and Mediterranean Sea, their optical properties being relatively close, we can estimate that ammonium represents about one third of the direct radiative forcing shown over the Euro-Mediterranean region (Figure 13 and 15). A sentence has been added to the article (section 5.1).

Last, it would help readability if equations are numbered within the manuscript.
Done.

---

## Editor Decision (ED1)

**Editor comments on the revised version of the ms. acp-2018-1101 by T. Drugé et al., entitled "Model simulation of ammonium and nitrate aerosols distribution in the Euro-Mediterranean region and their radiative and climatic effects over 1979-2016"**

François Dulac, 06 March 2019

Thank you for your corrections. I am pleased to accept your manuscript for publication in the ChArMEx Special Issue in ACP pending a few additional technical corrections listed hereafter (changes are underlined):

- Section 2. 1, p.4, line 17: Prospero et al., Environmental characterization of global sources of atmospheric soil dust…, Rev. Geophys., 2002, would be a good additional or alternative reference here.

- Section 2.2.3: p.8, l.7: Check "Another".

- Section 2.2.3: p.8, lines 24-25: I suggest to replace "be taken into account" by "stay in mind".

- Section 2.2.4: p.9, l.6: change to "close to the sulphate deposition velocity already used".

- Section 4.2, p.12, l.11: specify "(both Aqua and Terra)".

- References: use an index in "$NO_2$" in Georgoulias et al. (2018); insert Prospero et al., 2002 following above recommendation; capitalize the journal name "Climate Research" in Zanis et al. (2012).

---

## Author Response (AR2)

Editor comments on the revised version of the ms. acp-2018-1101 by T. Drugé et al., entitled "Model simulation of ammonium and nitrate aerosols distribution in the Euro-Mediterranean region and their radiative and climatic effects over 1979-2016"

**François Dulac, 06 March 2019**

Thank you for your corrections. I am pleased to accept your manuscript for publication in the ChArMEx Special Issue in ACP pending a few additional technical corrections listed hereafter (changes are underlined):
I would like to thank again François Dulac for his Co-Editor comments which mention various points listed below.

10    - Section 2. 1, p.4, line 17: Prospero et al., Environmental characterization of global sources of atmospheric soil dust..., Rev. Geophys., 2002, would be a good additional or alternative reference here.
The reference has been added.

- Section 2.2.3: p.8, l.7: Check "An another".
15    Done.

- Section 2.2.3: p.8, lines 24-25: I suggest to replace "be taken into account" by "stay in mind".
Done.

20    - Section 2.2.4: p.9, l.6: change to "close to the sulphate deposition velocity already used".
Done.

- Section 4.2, p.12, l.11: specify "(both Aqua and Terra)".
Done.

- References: use an index in "NO$_2$" in Georgoulias et al. (2018); insert Prospero et al., 2002 following above recommendation; capitalize the journal name "Climate Research" in Zanis et al. (2012).
Done.

[revised manuscript text omitted]